# Deep learning for rapid analysis of cell divisions in vivo during epithelial morphogenesis and repair

Jake Turley[1,2,3], Isaac V Chenchiah[1]*[†], Paul Martin[2]*[†], Tanniemola B Liverpool[1]*[†], Helen Weavers[2]*[†]

[1]School of Mathematics, University of Bristol, Bristol, United Kingdom; [2]School of Biochemistry, University of Bristol, Bristol, United Kingdom; [3]Mechanobiology Institute, National University of Singapore, Singapore, Singapore

*For correspondence:
Isaac.Chenchiah@bristol.ac.uk
(IVC);
paul.martin@bristol.ac.uk (PM);
t.liverpool@bristol.ac.uk (TBL);
helen.weavers@bristol.ac.uk
(HW)

†These authors contributed
equally to this work

Competing interest: The authors
declare that no competing
interests exist.

Reviewing Editor: Aleksandra
M Walczak, École Normale
Supérieure - PSL, France

## eLife assessment

In this **valuable** study, the authors use deep learning models to provide **solid** evidence that epithelial wounding triggers bursts of cell division at a characteristic distance away from the wound. The documentation provided by the authors should allow other scientists to readily apply these methods, which are particularly appropriate where unsupervised machine-learning algorithms have difficulties.

**Abstract** Cell division is fundamental to all healthy tissue growth, as well as being rate-limiting in the tissue repair response to wounding and during cancer progression. However, the role that cell divisions play in tissue growth is a collective one, requiring the integration of many individual cell division events. It is particularly difficult to accurately detect and quantify multiple features of large numbers of cell divisions (including their spatio-temporal synchronicity and orientation) over extended periods of time. It would thus be advantageous to perform such analyses in an automated fashion, which can naturally be enabled using deep learning. Hence, we develop a pipeline of deep learning models that accurately identify dividing cells in time-lapse movies of epithelial tissues in vivo. Our pipeline also determines their axis of division orientation, as well as their shape changes before and after division. This strategy enables us to analyse the dynamic profile of cell divisions within the *Drosophila* pupal wing epithelium, both as it undergoes developmental morphogenesis and as it repairs following laser wounding. We show that the division axis is biased according to lines of tissue tension and that wounding triggers a synchronised (but not oriented) burst of cell divisions back from the leading edge.

## Introduction

Significant advancements in confocal microscopy mean it is now possible to collect vast quantities of time-lapse imaging data from living tissues as they develop in vivo and respond to genetic or environmental perturbations (such as wounding). In parallel with the development of these imaging technologies, new methodologies are required to efficiently analyse these movies and extract detailed information about how the various cell behaviours (e.g., cell movements, divisions, etc.) contribute to tissue development and expansion, and how they enable repair responses following tissue damage (*Etournay et al., 2015*; *Nestor-Bergmann et al., 2019*; *Olenik et al., 2023*; *Park et al., 2017*; *Scarpa et al., 2018*; *Tetley et al., 2019*; *Turley et al., 2022*).

Computer vision (a form of artificial intelligence, AI) has progressed extensively in recent years, particularly with the development of deep learning models (*Guo et al., 2016*; *Voulodimos et al., 2018*). Such models can be trained to identify and classify objects in images, for example, enabling automated identification of tumours in MRI (Magnetic resonance imaging) scans or segmentation of cells according to their type (*Işın et al., 2016*; *Tran et al., 2018*). These algorithms are particularly useful when analysing medical and biological data (*Jones et al., 2017*), because these data are often inherently 'noisy', as objects within a class can exhibit significant variation.

Deep learning algorithms excel at finding patterns in complex data (*Guo et al., 2016*). These abstract patterns are often so complicated that they are difficult for the human eye to discern (*Bhatt et al., 2020*). In order to operate, deep learning algorithms must 'learn' from 'labelled data', that is, data in which an expert has already performed the task that we require the model to automate (e.g., segmentation, classification, etc.). Using this 'training' data – which includes both the input and correctly annotated output (ground truth) – the algorithm then learns how to complete the same task (*Howard and Gugger, 2020*). The resulting algorithms can be highly accurate at performing relatively simple vision tasks and are often much quicker than when the equivalent task is performed manually (*Jones et al., 2017*). This automated process allows efficient analyses of large datasets without extensive time-consuming repetitive work by a clinician or researcher. Furthermore, the high consistency of the resulting models reduces operator bias (or error) and can guarantee the same level of accuracy across all experiments and studies.

In microscopy, deep learning has, so far, largely been applied to static images and has enabled relatively simple analyses, such as cell counting and quantification of object area (or volume), as well as more sophisticated tasks, such as the capacity to distinguish different cell types (*Jones et al., 2017*), and for detection of mitotic indexes in biopsy tissue sections which have notoriously poor manual reproducibility (*Aubreville et al., 2020*; *Piansaddhayanaon et al., 2023*). AI approaches are increasingly employed in (and beginning to revolutionise) digital pathology (*Burlutskiy et al., 2020*; *Wang et al., 2022*), and whilst most current applications are to two-dimensional (2D) static images, there are opportunities for deep learning models to be applied to dynamic time-lapse videos.

The biological tissue we investigate – the *Drosophila* pupal epithelium – is densely packed with nuclei, and the developmental cell divisions are dispersed and rapid, each occurring over a period of only several minutes. Moreover, as with most fluorescently labelled live tissue, the pupal epithelium is somewhat prone to photo-bleaching, thus limiting the frequency at which sequential images can be collected whilst maintaining tissue health. All these factors need careful consideration as we attempt to develop a fully automatised algorithm to detect and analyse the divisions with a high degree of accuracy. We found that standard methods for tracking nuclei (such as TrackMate; *Tinevez et al., 2017*) failed to cope with the constraints of our in vivo imaging data and routinely confused epithelial cell divisions with migrating immune cells (that often contain nuclear material from engulfed apoptotic corpses), whilst also missing many mitotic events that are clear to the eye. However, whilst cell divisions in time-lapse movie data are often too dynamic for current methods for cell tracking, they do produce unique and reproducible patterns of motion. Hence, we turned to deep learning algorithms that have the power and flexibility to learn and subsequently detect these patterns.

Previous work on automated methods for detecting cell divisions has largely been performed on lower-density tissues with cells spread relatively far apart and mostly in vitro (*Gilad et al., 2019*; *Kitrungrotsakul et al., 2021*; *Nie et al., 2016*; *Phan et al., 2019*; *Shi et al., 2020*). Whilst considerable progress has been made over the years, many of these models are not accurate enough for biologists to analyse more complex in vivo data. Some existing approaches use unsupervised methods, which have the major advantage of not needing time-consuming hand labelling of data; however, these methods may struggle to cope with highly dense tissues and currently perform worse (i.e., exhibit a lower accuracy of detection) than supervised models (*Gilad et al., 2019*; *Phan et al., 2019*). This is likely to be particularly true after tissue wounding, where the algorithm needs to be able to accurately distinguish potential false positives (e.g., dynamic cellular debris or immune cells) from true mitotic events. One highly effective supervised method which is performed on low-density in vitro cells involves a series of three deep learning models: The first is a Low-Rank Matrix Recovery (LRMR) model that detects regions of interest where likely mitotic events occur (*Mao et al., 2019*). The next step involves classifying these videos to determine whether a division has occurred or not, using a Hierarchical Convolutional Neural Network (HCNN). Lastly, a Two-Stream Bidirectional

Long-Short Term Memory (TS-BLSTM) model determines the time in the video that the division occurred.

Here, we propose a simpler, direct single deep learning model that can detect cell divisions with high accuracy even with challenging conditions that require it to cope with very dense and dynamic (developing) tissues as well as with wound-induced debris. Despite these more challenging experimental imaging data, our relatively simple but highly effective model can complete the tasks well enough to be used to answer biological questions. This is achieved by using much deeper networks, based on highly effective and widely used image classifying models, and by increasing input information (e.g. using two independent fluorescent channels), which we show increases model accuracy. We have also developed a related deep learning model to compute the orientation of detected cell divisions.

Having established an effective mitosis detection model, we proceed to analyse cell divisions in time and space during epithelial morphogenesis and following wounding within living tissue in vivo. As expected, in the unwounded developing pupal epithelium, we observe that cell division density decreases linearly with time (*Etournay et al., 2015*). However, wounding triggers a synchronous burst of cell divisions at 100 min post-wounding, in a ring of epithelial tissue beginning several cell diameters back from the wound edge; this ring of proliferation becomes broader with increased wound size. In parallel, we have generated a related deep learning algorithm to determine the orientation of these cell divisions. We anticipate this deep learning algorithm could be widely applicable to the analysis of dynamic cell behaviours in a range of tissues that are amenable to study over extended time-courses, and, for such purposes, we have developed a publicly available plugin for use by others.

## Results

### A deep learning strategy efficiently identifies dividing epithelial cells in time-lapse imaging data

We chose to develop, and test the capability of, our model using the epithelium of the *Drosophila* pupal wing because of the optical translucency and genetic tractability of *Drosophila*, which makes it easy to generate tissues with fluorescently labelled nuclei and cell boundaries (*Etournay et al., 2015*; *George and Martin, 2022*; *Mao et al., 2011*). The *Drosophila* pupal wing epithelium undergoes extensive growth through rapid cell divisions early in pupal life (*Athilingam et al., 2021*; *Paci and Mao, 2021*), and can be imaged with high spatio-temporal resolution using live confocal microscopy. *Drosophila* pupae at 18 hr after puparium formation (APF) are removed from their brittle, opaque puparium to reveal the transparent pupal wing (*Weavers et al., 2018*; *Figure 1A*). The wing epithelium is a relatively flat 2D cell sheet, composed of two opposing cell layers, each one-cell thick. To analyse the cell behaviours involved in tissue repair, we use an ablation laser to generate sterile and reproducible wounds which heal rapidly within a few hours (*Weavers et al., 2016*). We further enhance reproducibility by localising our imaging and wounding to a particular region of the wing (*Figure 1B–D*).

To gather training data to build an algorithm that can reliably detect cell divisions, we performed time-lapse imaging of unwounded and wounded pupal wings, with each movie lasting 3 hr (*Figure 1E, F*). We generated a z-stack (with z-steps of 0.75 μm in depth) that encompassed the entire epithelial cell layer at each timepoint, which we then converted to a 2D image using a stack focuser tool (*Umorin, 2002*). For the wounded imaging data, the wounds generated possessed a mean radius of 20 μm (ranging from 9 to 30 μm), with the smallest wounds closing 20 min after wounding and the largest wounds taking up to 90 min to fully close. Crucially, tissue wounding created several imaging complications that our algorithm needed to accommodate. Firstly, wounding led to the epithelium around the wound edge moving out of the original focal plane which reduced the image quality at the immediate wound edge. This loss of image quality was further exacerbated by a wound-associated reduction in the levels of the Ecadherin-GFP junctional reporter (*Figure 1E, F*), which might be a consequence of the previously reported loosening of junctions in the migratory wound epithelium (*Martin and Nunan, 2015*; *Razzell et al., 2014*; *Tetley et al., 2019*). Secondly, wounding, by definition, leads to the accumulation of local tissue debris, including bright nuclear material. Motile immune cells and fat body cells, also with Histone2Av-mRFP-positive nuclei, are recruited to the wound and both of these cell lineages engulf tissue debris (*Franz et al., 2018*; *Razzell et al., 2011*); these motile

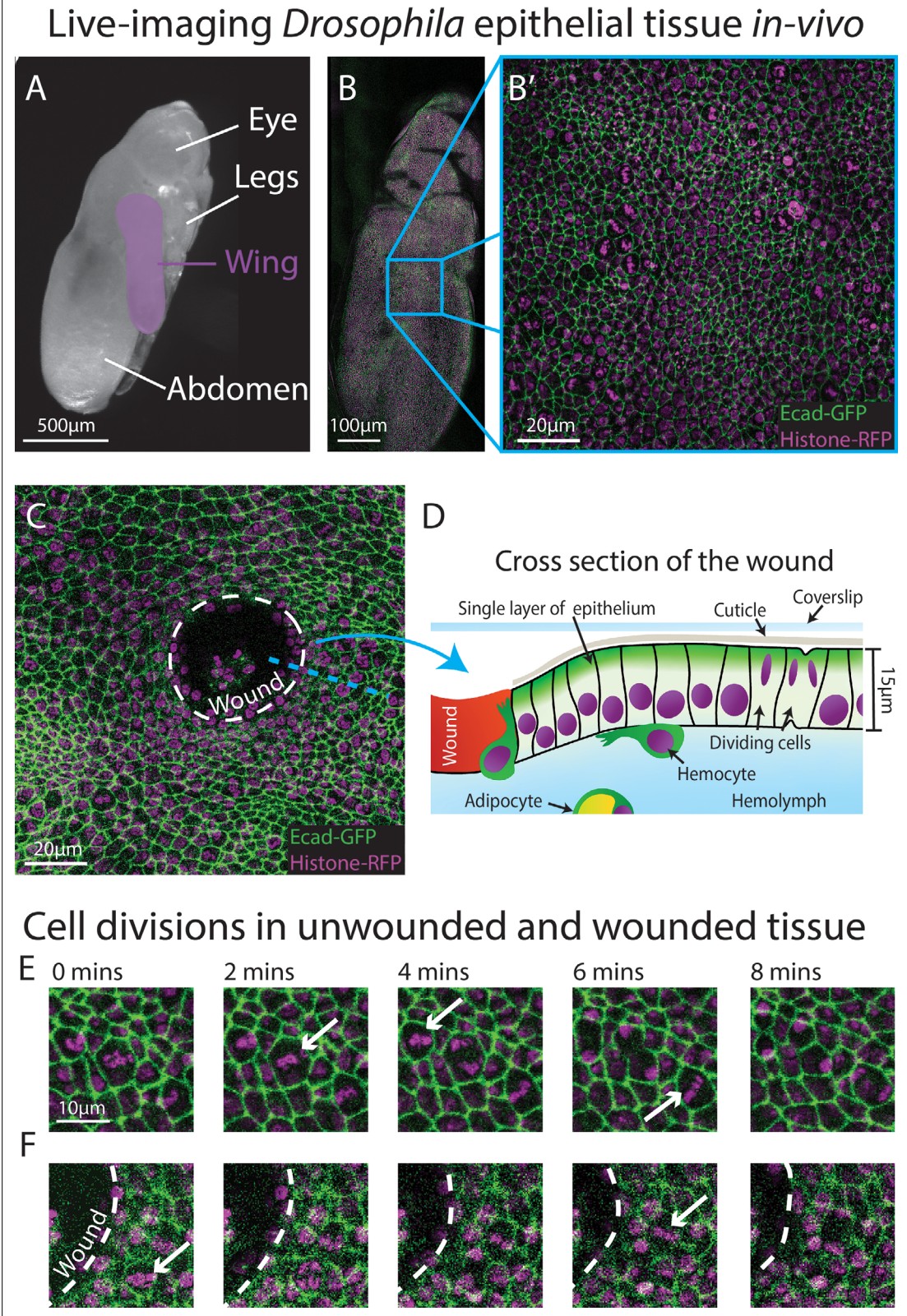

**Figure 1.** Live-imaging of *Drosophila* epithelial tissue dynamics in vivo. (**A**) Translucent *Drosophila* pupa with the pupal wing highlighted in magenta. (**B**) The pupal wing (with magnified inset, B', on the centre zone of the wing where we consistently image) with cell boundaries labelled using E-cadherin-GFP (green) and nuclei with Histone2Av-mRFP (magenta). (**C**) Magnified view of the pupal wing epithelium after wounding, with the white dashed line indicating the wound edge. (**D**) Schematic showing a cross-section through the upper layer of epithelium of the pupal wing, with

*Figure 1 continued on next page*

*Figure 1 continued*

haemolymph (insect blood containing haemocytes and adipocytes) beneath and rigid cuticle above (**E**) Multiple cell divisions (arrows) occur in the unwounded pupal wing epithelial tissue over the course of 8 min. (**F**) A cell division (arrow) occurs in a wounded epithelial tissue with the white dashed line indicating the wound edge.

and phagocytic (non-epithelial) cell types can be mistaken for dividing epithelial cells providing many opportunities for 'false positives'. Finally, since pupae are living, developing organisms, they occasionally (and unavoidably) move during imaging, leading to sample drift in-between frames, and this also leads to the generation of false positives.

To limit photo-bleaching of our biological samples, we chose to capture images every 2 min (*Figure 1E, F*), which affords the sample over 1 min of photo-recovery in between scans. Since the chromosomal separation phase of cell division (termed anaphase) takes approximately 6 min in this specific tissue, the chosen imaging frequency captures some details of each division, but is insufficient for the application of a standard (non-deep learning) algorithm. Particle tracking algorithms, which link nuclei together by the distance travelled, are also inappropriate here, since the pupal epithelial cells (and thus nuclei) are packed close together and dividing daughter nuclei would frequently (and mistakenly) be linked to a neighbouring nucleus rather than being associated with the parental cell. All these factors together make developing a highly accurate method to detect the vast number of cell divisions across our movies very challenging.

We have overcome these various image analysis constraints by generating a deep learning model to locate cell divisions in space and time from complex 2D+T imaging data (*Figure 2*; *Ji et al., 2013*; *Nie et al., 2016*; *Villars et al., 2023*). We use a ResNet34 model modified into a U-Net structure (*He et al., 2016*; *Ronneberger et al., 2015*). ResNet is a widely used architecture for red, green and blue (RGB) image classification. These deep learning models, with a convolutional neural network (CNN) architecture, are constructed of 'residual' layers (hence the name ResNet). Residual layers are specifically used to overcome the problem of degradation in which adding more layers makes optimising a model more difficult. These layers make it possible to construct networks with hundreds of convolutional layers, which allows deeper networks to be trained and thereby increases the networks' ability to accurately classify images (*He et al., 2016*). However, here we not only want to know *whether* a division has occurred in a given time period, but also to determine its *location* in space – and to do this we use a U-Net structure.

U-Nets were developed to segment images by classifying regions into categories. These neural networks have a U-shaped structure, with an encoder side that applies CNNs and other types of layers which decrease the spatial resolution whilst increasing features. The opposite happens on the decoder arm of the U-shaped structure, with the reintroduction of spatial information via skip connections allowing for the classification of individual pixels within the image (*Ronneberger et al., 2015*). In our system, we classify epithelial cells into 'dividing' or 'non-dividing' (the latter being the vast majority) and by their location in space. We envisioned a U-Net structured model based on a ResNet that will be able to classify far more accurately than the standard U-Net model. To boost the model's capacity to segment time-lapse videos, we used the fast.ai libraries Dynamic U-Net class which can create a U-Net structure from an image classifier (see Materials and methods for further details of the model architecture). This final model will therefore combine the properties of both models, enabling the training of high performing deeper networks with the U-Net structure. A key benefit of this method is that deeper/newer image classifier models can be swapped for more difficult tasks or to increase performance.

## Development of Deep Learning Model 1 (U-NetCellDivision3)

Both the standard ResNet and U-Net models use three channel RGB images as input. Here, our confocal z-stack images are composed of only two channels (E-cadherin-GFP, green, and Histone2Av-mRFP, red; *Figure 1E, F*), leaving a spare channel for other potential inputs. The clearest features of a dividing cell occur as the duplicated chromosomes separate and move in opposite directions (observed in the Histone2Av-mRFP channel, arrows, *Figure 2A*). Hence, we started developing our model by focussing only on the Histone2Av-mRFP channel, and use three sequential time-lapse images of the Histone2Av-mRFP (nuclear) channel (*Figure 2A*), the first frame being when the cell is still in metaphase (before chromosomal separation, $t = 0$ min) and the second and third in anaphase

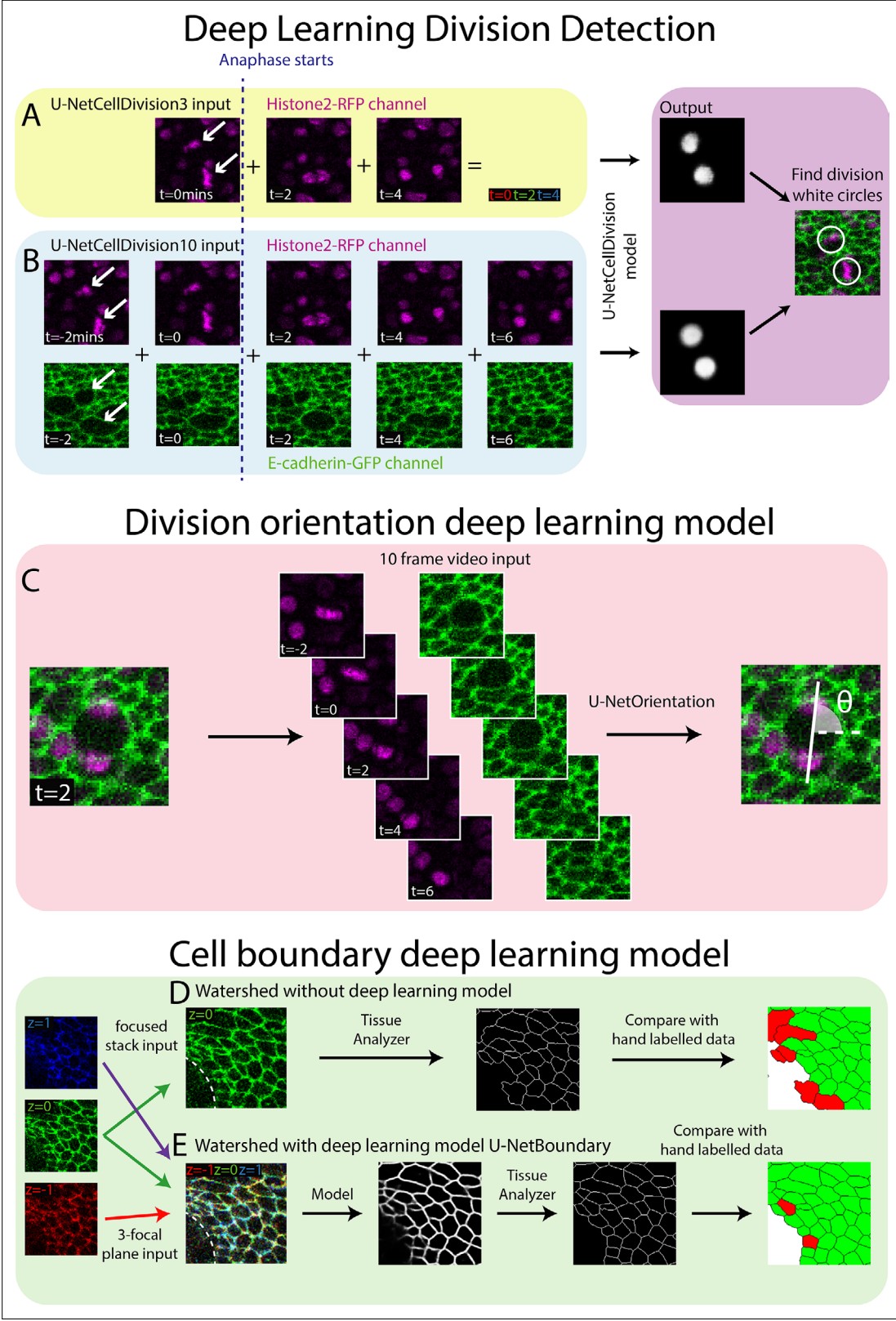

**Figure 2.** Deep learning detection of cell divisions, division orientation, and cell boundaries. Four deep learning models were developed to analyse epithelial cell behaviours. (**A**) The first version of the division detection model receives three frames from the Histone2Av-mRFP channel, which can be combined into a single RGB image, as is standard for a U-Net model. (**B**) The second version of the model input has 10 frames, 5 each from the Histone2Av-mRFP and E-cadherin-GFP channels. The model produces a white circle (white spot) wherever it detects a division. (**C**) The cell division

*Figure 2 continued on next page*

*Figure 2 continued*

locations are then passed through the U-NetOrientation model to determine the division orientation. This model takes 10 frames of a division as the input. (**D**) Segmentation of the focussed cell boundaries without using a deep learning model. The focussed stack image is inputted to Tissue Analyzer for segmentation and the result is compared to a hand-labelled ground truth. Green cells are correctly segmented and red cells are incorrectly segmented. (**E**) The three-focal plane image is inputted into the U-NetBoundary model and then segmented using Tissue Analyzer; this result is then compared to the hand-labelled ground truth.

(during and after chromosome separation, $t$ = 2 min and $t$ = 4 min, respectively). Representing these three sequential frames in different colours and combining them into a single RGB image reveals a clear pattern with broadly symmetric stripes of red (centrally) followed by green and blue (extending outwards) (*Figure 2A*). Crucially, there is a dramatic contrast between this triple-coloured division pattern and that of non-dividing cells that are relatively stationary and so appear as a white/grey circular shape (*Figure 2A*).

Our deep learning model is trained to distinguish between these different RGB patterns and thus to accurately detect and locate cell divisions. To train the model, we first manually identified dividing cells in 20 independent time-lapse videos of unwounded and wounded tissue (this generates 'labelled' training data); each training video consisted of 93 time frames (reflecting 186 min of footage). In this training data, we detected 4206 divisions in total across all movies. Next, we generated an 'output' that we required the model to be able to reproduce. For this, we generated a 'mask' video where every division was marked with a white circle (the same size as a cell about to divide) in the same location and at the correct time. The algorithm was then trained to reproduce this 'mask' (*Figure 2A*).

We trained this deep learning algorithm which we term 'U-NetCellDivision3'. Next, we tested the model on data it had not previously seen; the results are shown in *Table 1*; it should be noted that there are no experimental differences between each of the labelled datasets; they are comprised only of different biological repeats. The results (outputs) are categorised into (1) true positives ($T_p$) where a cell division has correctly been identified, (2) false positives ($F_p$) where the model has incorrectly detected a cell division where one has not occurred, and (3) false negatives ($F_n$) where a cell division occurred but the model failed to detect it. We can then compute 'Dice score' (*F*1 score) as a measure of the model's accuracy, by combining $T_p$, $F_p$, and $F_n$ (*Carass et al., 2020*). The dice score is defined as:

$$\text{Dice score} = \frac{2T_p}{2T_p + F_p + F_n}$$

A dice score of 1 is a perfect score, whereas scores progressively smaller than 1 indicate a poorer algorithm performance. Dice scores for our algorithm 'U-NetCellDivision3' indicate that this model detects only 78.7% of all cell divisions, and it led to many false positives (*Table 1*).

## Development of Deep Learning Model 2 (U-NetCellDivision10)

To overcome the false positives and negatives associated with our initial model, U-NetCellDivision3, we extended the deep learning model beyond a 3-frame input to increase the number of input frames to 10 (*Figure 2B*). Here, we included an additional timepoint either side of the original 3-frame images, taking our input data to 5 timepoints in total, and extended the analysis to include both the E-cadherin-GFP and Histone2Av-mRFP channels, thus incorporating the dynamics of both cell nuclei and cell boundaries. Consequently, two of these timepoints show the cell in metaphase and three timepoints show the cell moving through anaphase into telophase and cytokinesis (*Figure 2B*). Although there should be little nuclear movement in these first two frames, including these additional metaphase images will help filter out false positives due to dynamic non-mitotic nuclei. In this algorithm, to be identified as a dividing cell, the cell nuclei will need to be stationary in metaphase for

**Table 1.** Dice scores for the deep learning models.

| Model | True positives | False positive | False negative | Dice score |
|---|---|---|---|---|
| U-NetCellDivision3 | 797 | 216 | 310 | 0.752 |
| U-NetCellDivision10 | 1057 | 28 | 50 | 0.964 |

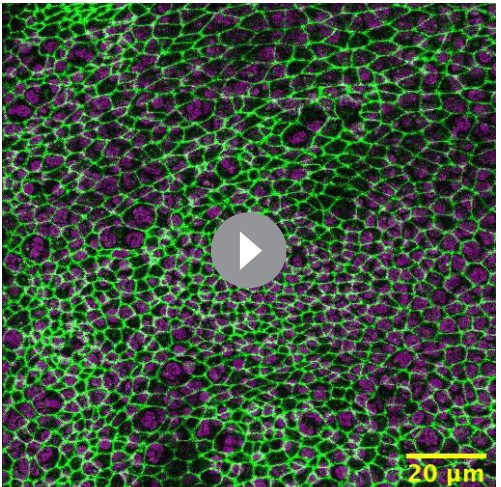

**Video 1.** Time-lapse imaging of the unwounded pupal epithelium over 3 hr. Projected from a 3D stack using the stack focus algorithm with a radius of 5 pixels. Green indicates E-cadherin-GFP and magenta indicates Histone2Av-mRFP. The white circles show the divisions detected by the 'U-NetCellDivision10' and the white lines indicate the orientation of divisions determined by 'U-NetOrientation'. Scale bar: 10 μm. Related to *Figure 3*.

https://elifesciences.org/articles/87949/figures#video1

two frames (2 min) before separating in a classical anaphase-like manner. Moreover, we included the E-cadherin-GFP channel to provide additional information on cell boundaries and further enable the model to identify dividing cells. Indeed, it is well documented that cell boundaries change prior to division as cells increase their size and become rounder (*Lancaster and Baum, 2014*), and indeed this can be observed in the pupal wing tissue (*Figure 1B and C* and *Figure 2B*). Inclusion of the E-cadherin-GFP channel should also help rule out false positives (such as nuclear debris within the wound centre), which will lack a clear GFP-positive boundary. Inclusion of the E-cadherin channel is particularly helpful in concert with the additional fifth timepoint, as the cells can be clearly observed to separate as they move through telophase and cytokinesis. A key finding of this study is that using multiple fluorescent channels can increase information about mitotic events which, in turn, leads to higher accuracy (fewer false positives and negatives).

We subsequently trained the model (Model 2) using the same data as previously used to train Model 1. As shown in *Table 1*, there is now a significant (over 80%) reduction in both false positive and false negatives using the 10-frame model. Most of the errors described previously have largely been resolved; a dice score above 0.95 means we can be far more confident in the results produced by U-NetCellDivision10. *Video 1* shows the cell divisions that the algorithm has correctly identified; the orientations of the divisions are also revealed (see later). Now, having established a deep learning algorithm that can accurately (and quickly) identify and quantify cell divisions from our in vivo imaging data, we used the model to explore how (and where) cell divisions occur within a living, developing epithelial tissue in normal conditions, and how this changes following an experimental perturbation such as wounding (*Figure 3* and *Figure 4*). We also later extended this strategy to develop additional deep learning models to study different aspects of cell behaviour (shapes of cell boundaries and identification of cell division orientation planes, *Figure 2C–E*).

## Cell divisions within unwounded epithelial tissue in vivo exhibit a 'community effect'

We first explored whether the 'U-NetCellDivision10' algorithm can be used to quantify the numbers and locations of cell divisions within the unwounded pupal wing epithelium of *Drosophila*. We initially used our algorithm to compute 'division density' over space and time, that is, the number of divisions occurring in a given area at a given time (*Figure 3A*). Interestingly, in the unwounded pupal epithelium, we observed that cells are more likely to divide close to and soon after previous divisions (*Figure 3B*). To explore this phenomenon and determine whether cell divisions occur randomly across the unwounded tissue or whether they are more likely to occur close to other divisions, we calculated a space–time correlation for the cell divisions (see Methods for details). The space–time correlation is shown as a heatmap (*Figure 3B*), with more intense red reflecting higher correlation. There is a high correlation close to the origin (within a 30-μm radius and temporally, within 40 min), which implies that cells are more likely to divide close to others in both space and time; this effect reduces with both increasing distance and time between cells. Consistent with previous studies of pupal wing morphogenesis (*Etournay et al., 2016*; *Milan et al., 1996*), we also find that the density of cell divisions decreased linearly with time during the developmental process (*Figure 3A*).

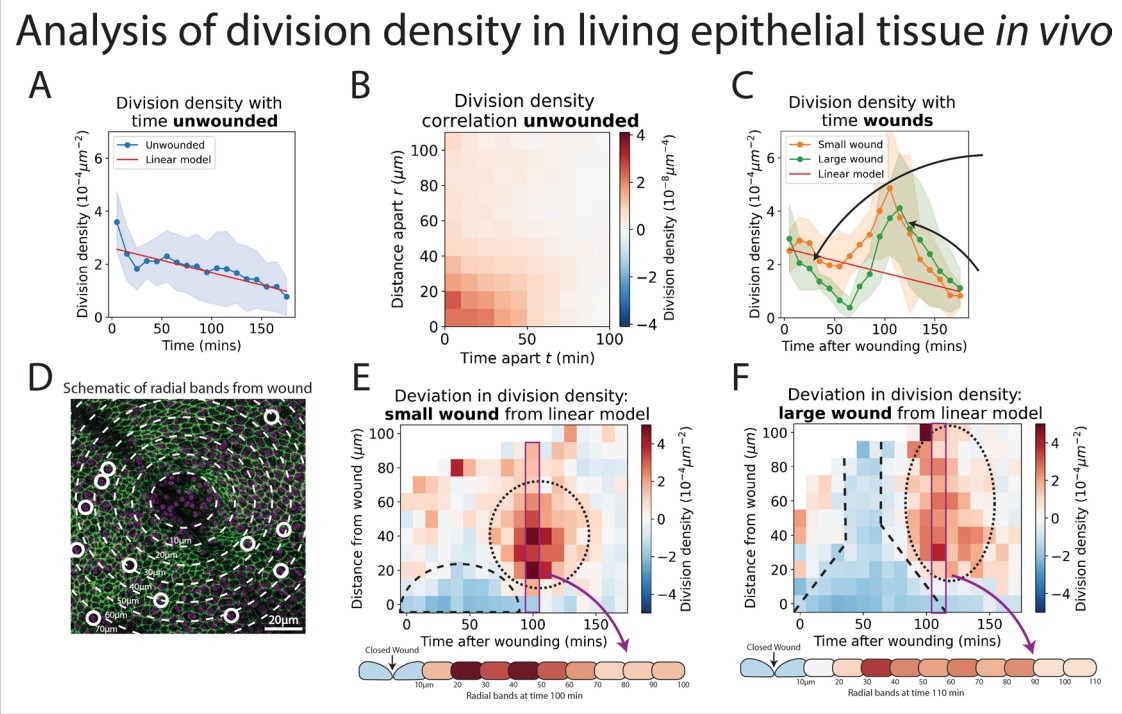

Analysis of division density in living epithelial tissue *in vivo*

**Figure 3.** Analysis of cell division density in living epithelial tissue in vivo. (**A**) The density of cell divisions in the unwounded tissue, with faded blue region showing the standard deviation. The red line is the line of 'best fit' of the unwounded data. (**B**) A heatmap of the division density correlation over distance and time in unwounded epithelial tissue. Red indicates positive, and blue negative correlation. (**C**) The density of cell divisions in the wounded tissue, with either small or large wounds, with faded regions showing associated standard deviation. The red line is the line of best fit of the unwounded data. The micrographs show representative divisions identified at two different timepoints post-wounding. (**D**) Diagram of the annular bands around a wound, each 10 m wide (white dashed line); white circles indicate cell divisions. (**E, F**) Heatmaps of the change in division density for small and large wounds compared with a best fit linear model of unwounded data. Red areas have more divisions, and blue less, than equivalent regions in the unwounded data. The dashed lines highlight areas in which cell divisions decrease and the dotted lines highlight areas in which divisions increase compared to unwounded data. Schematics below the heatmaps in E and F show the radial division densities 100 and 110 min after wounding, respectively (n = 14 unwounded, n = 8 small wounds, and n = 9 large wounds).

The online version of this article includes the following figure supplement(s) for figure 3:

**Figure supplement 1.** Further analysis of division density in living epithelial tissue.

## Epithelial wounding triggers a spatio-temporal reprogramming of cell division

Analysis of wounded tissues reveals striking differences in cell division behaviour when compared to unwounded tissue (compare *Figure 3A* with *Figure 3C*). These altered behaviours are highly dependent on the size of the wound. For larger wounds (15–20 µm radius), there are initially significantly fewer cell divisions (i.e., a lower division density) in the wounded epithelium compared to unwounded tissues (*Figure 3C*); this wound-associated inhibition of cell division reaches its low point at 60–70 min post-wounding. In contrast, smaller wounds (8–12 µm radius) do not exhibit a similar reduction in cell divisions immediately following wounding (*Figure 3C*). However, both small and large wounds exhibit a subsequent and dramatic synchronised burst of cell divisions at 100 min post-wounding, double that of unwounded tissue at the peak of this proliferative surge (*Figure 3C*); after 3 hr post-wounding, the division density of wounded tissue returns to unwounded levels (*Figure 3C*). We calculated the space–time correlation for the cell divisions in the wounded tissue and found a similar high spatial correlation around the origin with the same range as unwounded tissue (*Figure 3—figure supplement 1B, C*); nevertheless, the temporal correlation was altered, due to the observed suppression and later synchronisation of divisions caused by wounding.

Since our model also identifies the spatial coordinates of the cell divisions, we can determine their distance from the wound edge, and this enables us to calculate the density of divisions in zones extending out from the wound (*Figure 3D*). To analyse how the wounded division density varies

# Analysis of division orientation in living epithelial tissue *in vivo*

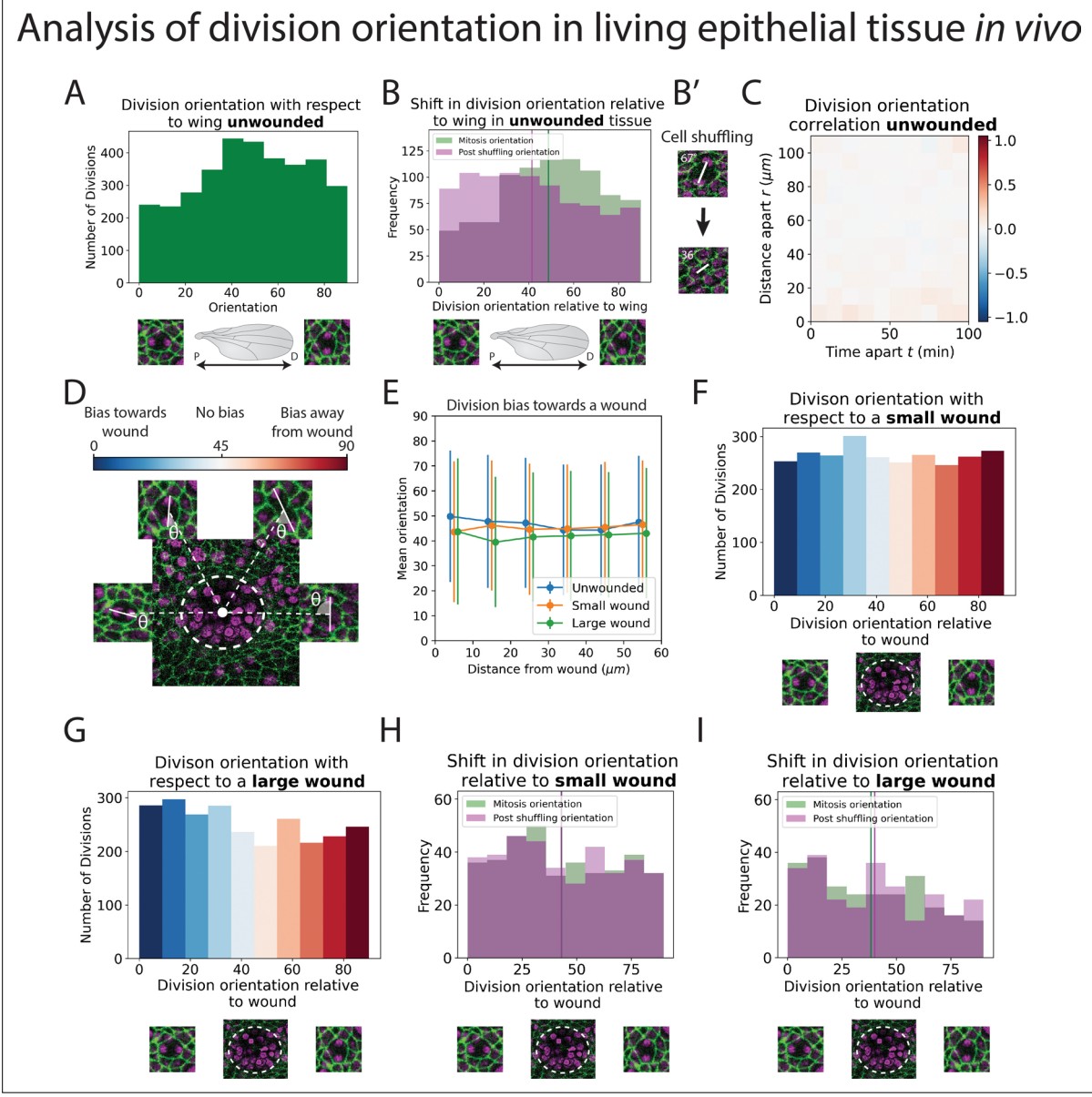

**Figure 4.** Analysis of division orientation in living epithelial tissue in vivo. (**A**) Distribution of the division orientations with respect to the proximal–distal axis of the pupal wing in unwounded tissue. Cell division orientations of 0° and 90° are illustrated in the micrographs. (**B**) Distribution of the division orientations with respect to the wing in unwounded tissue (green) and the daughter cell orientations 20 min after dividing (magenta), with examples of the orientation of division before and after cell shuffling (**B'**). (**C**) Heatmap of the space–time correlation of division orientation. Red indicates positive correlation, blue negative, and white no correlation. (**D**) Diagram of cell division orientation with respect to a wound; lower values are dividing towards the wound and higher values away. (**E**) Mean division orientation towards the wound as a function of distance from wound for small and large wounds. For unwounded tissues an arbitrary point is chosen as a 'virtual wound'. (**F, G**) Distribution of the division orientations with respect to small and large wounds. The spectrum of colours (same as in D) indicates the bias in orientation towards the wound. (**H, I**) Distribution of the division orientations with respect to the wound in small and large wounds (green), and the daughter cell orientation 20 min after dividing (magenta) (n = 14 unwounded, n = 8 small wounds, and n = 9 large wounds).

The online version of this article includes the following figure supplement(s) for figure 4:

**Figure supplement 1.** UNetOrientation model diagram and error of test data.

**Figure supplement 2.** Further analysis of division orientation in living epithelial tissue.

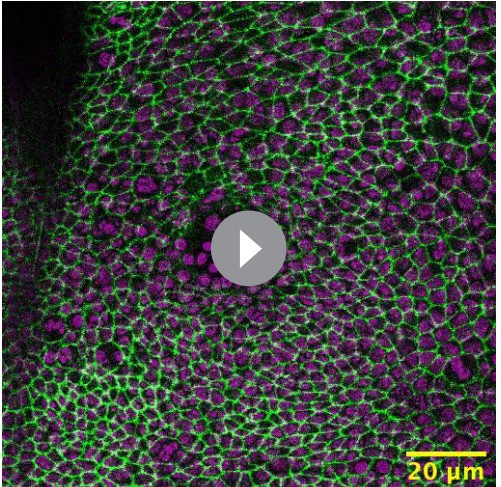

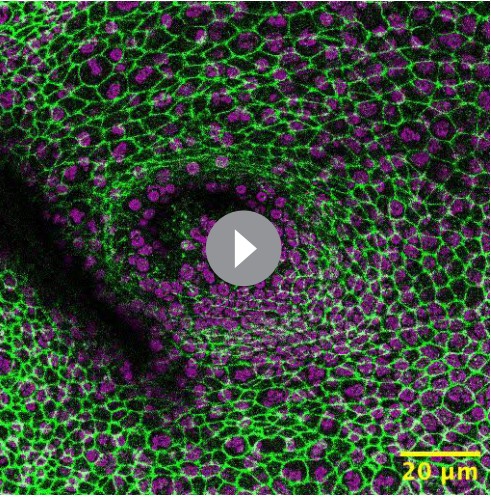

**Video 2.** Time-lapse imaging of a small wound in the pupal epithelium over 3 hr. Projected from a 3D stack using the stack focus algorithm with a radius of 5 pixels. Green indicates E-cadherin-GFP and magenta indicates Histone2Av-mRFP. The white circles show the divisions detected by the 'U-NetCellDivision10' and the white lines indicate the orientation of divisions determined by 'U-NetOrientation'. Scale bar: 10 μm. Related to *Figure 3*.

https://elifesciences.org/articles/87949/figures#video2

**Video 3.** Time-lapse imaging of a large wound in the pupal epithelium over 3 hr. Projected from a 3D stack using the stack focus algorithm with a radius of 5 pixels. Green indicates E-cadherin-GFP and magenta indicates Histone2Av-mRFP. The white circles show the divisions detected by the 'U-NetCellDivision10' and the white lines indicate the orientation of divisions determined by 'U-NetOrientation'. Scale bar: 10 μm. Related to *Figure 3*.

https://elifesciences.org/articles/87949/figures#video3

over space and time, we have compared the wounded division data to that of unwounded tissue (by making a line of best fit for the unwounded data as a linear model and comparing the wounded data to this). This enables us to show the spatial-temporal change in division density in a heatmap, with blue indicating a decrease and red an increase in division density (*Figure 3E, F*). For small wounds, there is a clear decrease in divisions extending up to 20 μm (approximately 5-cell diameters) back from the wound edge until 70 min post-wounding. In large wounds, this reduction in division density extends much further back from the wound edge, beyond even the field of view (i.e., greater than 100 μm, approximately 25-cell diameters). The subsequent synchronised burst of divisions occurs between 20 and 70 μm back from the edge of small wounds and extends beyond 100 μm across the whole field of view for large wounds (*Videos 2 and 3*).

## The orientation of cell divisions might be biased by tissue tension but is not influenced by wounding

In addition to a general analysis of cell division density, we can also use our models to quantify the orientation of cell divisions in an automated manner. To achieve this, we developed a second deep learning model called 'U-NetOrientation'. Whilst our earlier model reveals the locations of dividing cells, we retrained this algorithm to report division *orientation* using nuclear positioning. To achieve this, we used the same model architecture as U-NetCellDivision but retrained it to complete this new task. Our new workflow first uses U-NetCellDivision10 to find cell divisions. Secondly, U-NetOrientation is applied locally to determine the division orientation. The same cell divisions from the previous training videos were used to train the U-NetCellDivision model. We initially labelled the cell division orientations by hand and then trained the new deep learning model to extract $\theta$, that is, the orientation of the division (see *Figure 4—figure supplement 1A*). After training, we tested the model's accuracy by comparing the hand-labelled orientation with the one from the model. We found that the median difference between these values was $4°$ ($\pi/45$ *radians*) (*Figure 4—figure supplement 1B*; *Videos 1–3*).

Following this model training and validation, we used the U-NetCellDivision model to quantify division orientation in unwounded and wounded epithelial tissues (*Figure 4*). In the unwounded pupal epithelium, we measured the division orientation relative to the proximal/distal (P/D) axis of the wing

**Table 2.** Dice scores for the segmentation methods.

| Segmentation | True positives | False positive | False negative | Dice score |
|---|---|---|---|---|
| Single focal plane + Tissue Analyzer | 8197 | 313 | 4317 | 0.780 |
| U-NetBoundary + Tissue Analyzer | 11,325 | 501 | 1189 | 0.931 |

(*Figure 4A*). Previous work has demonstrated that hinge contraction in the proximal part of the wing causes tension, resulting in cells becoming elongated in the wing along the P/D axis (*Athilingam et al., 2021*; *Etournay et al., 2016*) and because of this, we anticipated that a bias of division orientation might occur along this axis. However, surprisingly, we observe a small orientation bias at 45° to this axis (*Figure 4A*). Interestingly, our subsequent analysis revealed that daughter cells undergo later 'shuffling' movements to rearrange their positions after cytokinesis so that the *final* daughter cell orientations (using centres of the cell shapes) consistently align along with the P/D axis (*Figure 4B*). To analyse these 'shuffling' rearrangements, we needed to segment and track cell boundaries. However, applying traditional tools, such as the ImageJ Tissue Analyzer plugin (*Etournay et al., 2016*), we found that our samples were too noisy to analyse without time-consuming manual editing of the segmentation. Hence, we automated this cell boundary segmentation by developing an additional (fourth) deep learning model to detect cell boundaries (*Figure 2D*; *Abedalla et al., 2021*; *Aigouy et al., 2020*; *Fernandez-Gonzalez et al., 2022*; *Wolny et al., 2020*). Here, we developed a model using multiple focal planes (individual slices of the z-stack) from the confocal imaging data. This allowed us to take advantage of the fact that E-cadherin is visible in the top few (2–3) z-slices of cells. Using this 3D data gives a clear signal in this otherwise noisy wounded tissue data (*Wolny et al., 2020*). We therefore used a three-focal plane input to increase the amount of information available to the model, which we have called U-NetBoundary (*Figure 2D, E*). This utilised an algorithm which identifies the most focussed plane, and the planes immediately above and below it (see Materials and methods for further details), to provide sufficiently accurate automated identification of cell boundaries. After training, we tested the U-NetBoundary model on 12 images (12,514 cells) and ran the output through ImageJ's Tissue Analyzer (*Etournay et al., 2016*) to utilise the Watershed algorithm. *Table 2* shows that using U-NetBoundary leads to a much better dice score and so is more reliable than using a single focal plane without deep learning.

Using U-NetBoundary, we are now able to track daughter cells post-division, with the required level of accuracy. Our algorithm automatically filters out any tracks that have large and/or sudden changes in size and shape of daughter cells, which indicates a likely segmentation mistake (see Methods for details). Once these anomalies have been identified and removed, our data is ready for analysis. To determine whether daughter cell orientation relative to one another changed during cell shuffling (in the period immediately after dividing), we measured the angle of a line drawn between the centres of two daughter nuclei 20 min after cell division (*Figure 4B'*) to find the change in orientation. We found that, on average, post-division shuffling

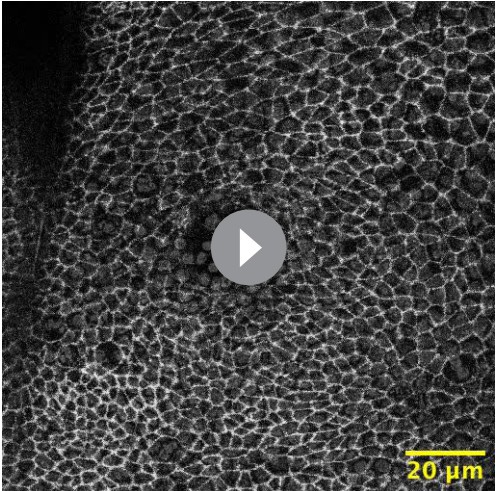

**Video 4.** Time-lapse imaging of a small wound in the pupal epithelium over 3 hr. Projected from a 3D stack using the stack focus algorithm with a radius of 5 pixels. Greyscale background of epithelium with circles show the divisions detected by the 'U-NetCellDivision10', the lines indicate the orientation of divisions determined by 'U-NetOrientation' and the colour of labels display the orientations relative to wounds. Blue labelled divisions are orientated towards wounds, red away from wounds and white around 45°. The white dot is the centre of the wound and the closed wound site after closure. Scale bar: 10 µm. Related to *Figure 3*.
https://elifesciences.org/articles/87949/figures#video4

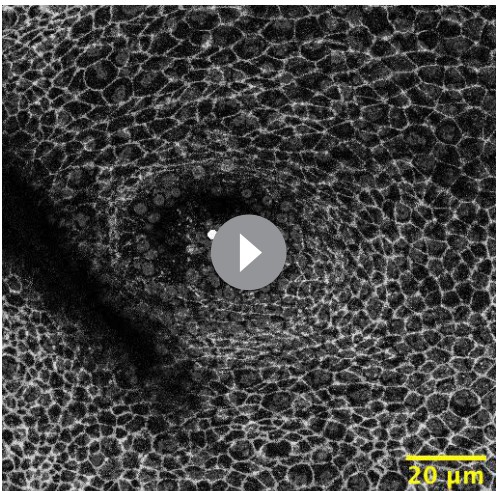

**Video 5.** Time-lapse imaging of a large wound in the pupal epithelium over 3 hr. Projected from a 3D stack using the stack focus algorithm with a radius of 5 pixels. Greyscale background of epithelium with circles show the divisions detected by the 'U-NetCellDivision10', the lines indicate the orientation of divisions determined by 'U-NetOrientation' and the colour of labels display the orientations relative to wounds. Blue labelled divisions are orientated towards wounds, red away from wounds and white around 45°. The white dot is the centre of the wound and the closed wound site after closure. Scale bar: 20 µm. Related to *Figure 3*.

https://elifesciences.org/articles/87949/figures#video5

shifted final daughter cell orientations by 14.8°. When we measured the post-shuffling orientation relative to the wing's P/D axis, we found that the distribution had shifted to acquire a small bias in the direction of the tension in the tissue (*Figure 4B*), and the mean orientation relative to the P/D axis had shifted by 8.5° to align with the global tension. If cell division orientation is also influenced by local tension in a developing tissue, then one might expect that cells about to divide in close proximity to one another will experience similar local forces and so might divide with the same orientation. To examine if this was the case, we measured the correlation between division orientation within space and time, but found no such correlation (*Figure 4C*). Therefore, we conclude that (1) global tension has a small influence on division orientation, but only after shuffling (repositioning) of the daughter cells and (2) local tension from the wound does not have a detectable effect. Other predictors of division orientation, such as cell shape, could dominate (*Nestor-Bergmann et al., 2019*).

Next, we measured division orientation relative to a wound (*Figure 4D–G*). Here, the possible range of cell division orientations varies from 0° to 90°, with an orientation of 0° indicating that cells divide towards a wound (radially), and an orientation of 90° indicating that cells divide perpendicular to a wound (tangentially). To investigate whether cell division is biased towards the wound, we averaged the orientation of all divisions around the wound. If divisions are biased towards a wound, then their average orientation should be significantly less than 45° (or above 45° if significantly biased away from the wound); conversely, an average bias of 45° would suggest that cells divide in an unbiased manner. From the uniform distribution of division orientations, our data suggest that a rather unbiased orientation of cell divisions occurs in response to wounding (*Figure 4D–G*, *Videos 4 and 5*). Whilst these data suggest that there is no initial bias in the orientation of cell divisions in the epithelium following wounding, we wondered whether subsequent 'shuffling' of daughter cells might be influenced by tissue tensions within the repairing epithelium. We undertook the same tracking of daughter cell movements as described for unwounded tissue (*Figure 4B*), but observed no significant shift in the cell orientations post-shuffling; rather, the distribution of post-division orientations is the same as for the divisions themselves (*Figure 4H, I*), suggesting that the local tension changes triggered by wound healing are not sufficient to have a measurable effect on the orientation of cell divisions, over and above those seen in unwounded tissue (*Figure 4—figure supplement 2*).

## Discussion

Deep learning is well suited to detecting variable but specific features in a dense field, such as face detection in a crowd. Hence, it is particularly useful for identifying patterns in noisy biological or clinical data. A key step with these data is identifying inputs for a given task, and this will be somewhat bespoke and dependent on the type of data being analysed. Here, we analyse confocal microscopy movies of translucent three-dimensional epithelial tissue, to identify and classify cellular behaviours in space and time during morphogenesis and tissue repair. To this end, we have developed deep learning tools for identifying and locating when and where cell divisions occur, as well as their division orientation and the post-division shuffling behaviour of daughter cells in unwounded and wounded tissue. This has allowed us to ask quite nuanced questions about cell division behaviours across an epithelial

field, as well as investigate how an individual cell division might influence local cell behaviours by its close neighbours.

For such dynamic cell behaviours as cell division, there is a clear need to analyse imaging data from high-resolution confocal microscopy movies of living tissue. Because of the vast volume of this data, doing this task manually would not be possible, and so one must develop sophisticated deep learning strategies for the analysis. Our approach has been to generalise techniques currently used in computer vision for static images and adapt them to deal with dynamic data. Previous deep learning approaches have considerably improved the detection accuracy of mitotic indexes in static biopsy sections of clinical tissues for cancer diagnosis (*Aubreville et al., 2020*; *Piansaddhayanaon et al., 2023*). We have built on other existing 3D CNN networks (*Ji et al., 2013*; *Nie et al., 2016*) by making deeper models which can receive multiple florescent channels. In our study, we successfully applied this type of analysis to very dense in vivo tissues which are undergoing the highly dynamic events involved in tissue development and repair following wounding. Despite these additional difficulties, our model proved to be highly accurate. Furthermore, we could also determine the orientation of cell divisions.

## What are the biological findings so far?

Our deep learning tools have enabled us to accurately quantify complex cell behaviours – such as cell divisions and subsequent daughter cell rearrangements – from large datasets which, in turn, has revealed trends that are otherwise hidden within the biology. Previous studies of wound healing in mammalian models have suggested that cell migration and cell division largely occur in separate epidermal domains following wounding (*Aragona et al., 2017*; *Park et al., 2017*) and our data support this. Our large wounds show a clear reduction in cell divisions, below pre-wound levels, in cells close to the leading epidermal wound edge where cells are actively migrating. Nevertheless, our data suggest that cell migration is not absolutely dependent on this 'shut down' in divisions because we see no observable cessation of cell division around small wounds as they are closing. For both large and small wounds, we observe a synchronised proliferative surge of cell divisions commencing 60 min post-wounding (and peaking shortly afterwards), but this is restricted to a domain beginning about 5-cell diameters back from the leading edge. These divisions are unlikely to be a major driver of wound closure because the rate of wound closure is the same before and after the proliferative surge. Indeed, cell divisions at the leading edge have largely halted during the most dramatic closure period. However, these cell divisions are likely to be a consequence of wounding, and the additional cells will help repopulate the tissue to restore near original numbers of epithelial cells and return tissue structure (and tensions) to pre-wound levels. This synchronised surge in cell proliferation in a band of cells back from the leading edge (to levels that are twice background levels for unwounded tissue) is potentially related to our observation of a strong correlation in the timing of cell divisions by close neighbours in unwounded epithelial tissue. Such a 'community effect' might be mediated by short-range chemical signals or local mechanical signals that operate locally in unwounded tissues and are recapitulated and expanded following wounding.

Once a cell has received signals driving it to divide, how do tissue tensions influence the orientation of this cell division in the unwounded or wounded epithelium? Previous studies of cells adjacent to the segmental boundaries in the *Drosophila* embryo show how local tissue tensions, driven by contractile actomyosin cables, can orient the plane of cell divisions adjacent to these boundaries (*Scarpa et al., 2018*). Moreover, analyses of experimentally stretched Xenopus tissue revealed that whilst global cell division rates are regulated by tissue-level mechanical stress, division orientation is controlled more locally by cell shape (*Nestor-Bergmann et al., 2019*). In our studies, we observe cells dividing with no specific orientation bias along the global P/D axis; however, subsequently, we do see the resulting daughter cells shuffle to adopt an alignment more biased along this P/D tension axis. We see no apparent bias in orientation of cell divisions following wounding; this was unexpected as one might presume there to be considerable tissue tension changes in the vicinity of a wound (*Guzmán-Herrera and Mao, 2020*; *Scarpa et al., 2018*). However, this effect might be partially explained by our observation that most cell divisions are distant from the main source of changing wound tensions, the contractile actomyosin purse-string that rapidly assembles in the leading epithelial wound edge cells (*Tetley et al., 2019*; *Wood et al., 2002*), and that these divisions occur largely after the wound has closed.

To further extend these studies and to gain a more comprehensive understanding of how different cell behaviours, particularly beyond those directly related to cell division, coordinate in a repairing tissue, additional development of our deep learning algorithms might be useful to extract more information from the time-lapse imaging data. For example, this might enable us to correlate changes in the density or orientation of cell divisions at the wound site, with other contributing cell behaviours (such as cell shape changes and cell intercalations). Similarly, it would be possible to integrate our analyses of cell behaviour with tools that enable live-imaging of wound-induced signalling (e.g., calcium signalling at the wound edge using calcium sensitive GCaMP transgenic reporters), in order to determine how such signalling pathways might be integrating the various wound repair cell behaviours following injury.

## Future directions for our deep learning approaches

In this study, we have converted a suite of image classifiers (ResNets) into U-Net via the Dynamic UNET function from the fast.ai library. To analyse cell divisions in *Drosophila* pupal tissues we extended the dimension of the data being analysed to include multiple time steps to identify the dynamic features associated with individual cell division events. To achieve this, we have modified the architecture of these models by increasing the feature inputs in the first layer. With tweaks, the model can provide us with additional, but related, outputs, for example, detection of defective cell divisions, which might be relevant in studies of oogenesis or cancer. Our algorithms could also be extended further by altering the initial layers of the model; this would enable the generation of models which can identify much more complex dynamical features. Indeed, a major challenge is to generate AI (or deep learning) models that can be adapted to identify cellular (or pathological) features across a broad range of tissue types and in data generated through a range of different imaging modalities. The tissue employed in our current study was a relatively flat 3D epithelium with few other cell lineages present in the microscopy data (only migratory immune cells), but such AI approaches could be expanded to cater for mixed populations of cells existing in larger 3D volumes, for example, gastruloids or even whole embryos as they develop and undergo morphogenesis, or to study other complex cell:cell interactions or movements, for example, immune cell interactions or flagella beating. Incorporating LSTM architectures could also help detect these dynamic and complex behaviours (*Mao et al., 2011*; *Shi et al., 2020*). With any such methodology, there will be much interesting work to come, in optimising movie time resolution, fluorescent markers and model depth.

The development of the next generation of *realistic* theoretical models of tissue mechanics during morphogenesis and repair (and other physiological episodes such as cancer progression) in vivo, will require dealing with increasingly large and complex imaging datasets. To extract information from them will require the use of further deep learning tools to automate the process of data extraction (of, e.g., cell velocities, cell shapes, and cell divisions). The theories that must be developed will be founded on non-equilibrium statistical mechanics applied to describing stochastic equations of motion for many microscopic interacting degrees of freedom. Identifying the most important features of the dynamics and quantifying the fluctuations will be highly challenging. We envision promising approaches will include (1) inferring equations of motion based on optimising the parameters of Partial Differential Equations (PDEs) for continuum fields (e.g., nematic director fields for cell orientations) using deep learning (*Colen et al., 2021*) or (2) reverse engineering the dynamics based on spatio-temporal correlation functions (of, e.g., cell shapes and cell velocities) that deep learning tools can elucidate. An advantage of the second approach is that one can also estimate the scale of fluctuations in the system.

The deep learning models that we present here can identify cell divisions and their orientations (as well as subsequent orientations of daughter cells) in dynamic movie data with high accuracy. We anticipate our models will have broad application and enable a similar analysis of tissues where cell nuclei and/or membranes have been labelled. To facilitate this, we have made a napari plugin that can run our model to detect cell divisions, which can be found at https://github.com/turleyjm/cell-division-dl-plugin (copy archived at *Turley, 2024*). All the data used to train/test the models plus the additional data used in the analysis cell divisions in wounded tissues can be found on our Zenodo dataset https://zenodo.org/records/10846684. Other researchers may wish to use different numbers of timepoints and/or fluorescent channels, leading to modifications in the number of frames inputted into the model; we provide clear instructions on how to do this in the code. Ultimately, we envisage

that such deep learning approaches are an important step towards the development of AI tools for analysing dynamic cell behaviours, including cell divisions, in complex physiological as well as pathological processes occurring in a variety of organisms and tissue types.

## Materials and methods

### *Drosophila* stocks and husbandry

*Drosophila* stocks were raised and maintained on Iberian food according to standard protocols (*Greenspan, 1997*). All crosses were performed at 25°C. The following *Drosophila* stocks were used: *E-cadherin-GFP* and *Histone2Av-mRFP* (BDSC stock numbers #60584 and #23651, respectively, obtained from the Bloomington *Drosophila* Stock Centre, Indiana).

### Confocal imaging and data processing

*Drosophila* pupae were aged to 18 hr APF at 25°C. Dissection, imaging, and wounding were all performed as previously described (*Weavers et al., 2018*). The time-lapse movies were generated using a SP8 Leica confocal. Each z-stack slice consisted of a 123.26 × 123.26 µm image (512 × 512 pixels) with a slice taken every 0.75 µm. The z-stacks were converted to 2D using our own Python version of the ImageJ plugin stack focuser (*Umorin, 2002*). Images were taken every 2 min for 93 timepoints (just over 3 hr of imaging). The data were manually labelled by making a database of the locations in space and time of the divisions and their orientations.

From the 93-frame full videos, we extracted 5 sequential timepoints to make a video clip (10 frames encompassing 2 different channels) or 3 timepoints and frames (for the U-NetCellDivision3). This was performed 89 times (one for each timepoint, apart from the last four timepoints). Our training data thus consisted of 979 video clips (11 full videos), our validation data consisted of 356 video clips (4 full videos) and our testing data consisted of 445 video clips (5 full videos). There are 4206 divisions across all the videos (on average, there are 2.38 divisions in each clip). There is no experimental difference between each of the labelled datasets as they are comprised of different biological repeats. Training data is used to directly teach the model to perform its given task. Validation data is used during the training process to determine whether the model is overfitting. If the model is performing well on the training data but not on the validating data, then this is a key signal that the model is overfitting and changes will need to be made to the network/training method to prevent this. The testing data is used after all the training has been completed and is used to test the performance of the model on fresh data it has not been trained on. We define the time a division occurred as the last timepoint of metaphase before anaphase starts. For each clip, we make a corresponding output mask (also 512 × 512 pixels) with divisions labelled with a white circle. This is generated using our hand-labelled database, which has the information about each division's location in space and time. These video clips (plus their corresponding output masks) are the labelled data that will be used for training the U-NetCellDivision deep learning models.

For calculating the orientation of cell divisions, we used the 10-frame video clips. For each division, we made a cropped video clip that was a 14.4 × 14.4 µm square box around the centre of a division. The images in the cropped video were 60 × 60 pixels (which we rescaled to 120 × 120 as this improved the performance of the models). The same training dataset that was used for training the U-NetCellDivision models wasz used for U-NetOrientation, with 2638 cropped video clips from the 11 full videos. The validation data was 660 cropped video clips from 4 full videos, and testing had 1135 cropped clips from 5 full videos. The output for the model is an oval elongated in the same orientation as the division. The oval has a radius of 50 pixels in the long axis and 15 in the short axis. In the labelled data, each division's orientation was measured by hand and the corresponding oval mask was generated. The mask is also 120 × 120 pixels.

For detecting cell boundaries, we maximise the information supplied to the model by using a modified stack focuser which identifies the most 'in focus' pixels in a stack. Our version also outputs the pixels above and below the most in-focus pixel and records this as an RGB image with colours corresponding to above (R), focussed (G), and below (B) pixels; the model will learn to use these upper and lower colours to identify if there is a genuine cell boundary or if focussed pixels are just noise within the image. We also rescaled our images from 512 × 512 pixels to 1024 × 1024 pixels, to increase the width of the boundaries so that they are large enough for the model to learn to detect them. The

data was segmented using Tissue Analyzer to apply the Watershed algorithm on the original single focal plane data (then correcting by hand the boundaries on 59 images, finding a total of 58,582 cells). The boundaries are 1 pixel in width in the output labelled masks. To give the model a wider target to reproduce, we eroded the image to make the boundaries 3 pixels wide. As we have increased the scale of the images, this is around the same pixel thickness as the boundaries in the input.

## Network architecture

We converted a Resnet34 model into a U-Net architecture via the Dynamic UNET function from the fast.ai library (*He et al., 2016*; *Howard, 2018*; *Ronneberger et al., 2015*). The weights from the Resnet 34 classifier were used to take advantage of transfer learning (*Howard and Gugger, 2020*). For the second version of the model (U-NetCellDivision10), the first layer of the model was replaced with a Conv2d layer with 10 features in and 64 out. The inputs to the model were 512 × 512 × 3 or 10 × 512 × 512 voxels for U-NetCellDivision3 or U-NetCellDivision10, respectively. U-NetCellDivision3 has 41405589 parameters and both U-NetCellDivision10 and U-NetOrientation have 41,268,871, all have 54 layers. The U-NetOrientation has the same architecture as U-NetCellDivision10, but takes 10 × 120 × 120 videos as inputs. For U-NetBoundary we used the Resnet 101 classifier and converted it into a U-Net with Dynamic UNET function. U-NetBoundary has 318,616,725 parameters and has 121 layers. This model has inputs of 1024 × 1024 × 3. Source code is available at https://github.com/turleyjm/cell-division-dl-plugin (copy archived at *Turley, 2024*).

## Data augmentation

The data were augmented using the albumentations library (*Buslaev et al., 2020*). The transforms used were Rotate, HorizontalFlip, VerticalFlip, GaussianBlur, RandomBrightnessContrast, and Sharpen.

## Training models

Training our deep learning models requires that we split the data into three separate groups (*Howard and Gugger, 2020*): (1) Training data: this is data from which the model directly learns (in this instance, the 11 videos described above); (2) Validation data: this data is used to test the model during the training process, to validate whether the algorithm is learning the patterns correctly and if it can perform on (similar but) unfamiliar videos. This ensures that the model has not simply 'remembered' the 'answer' in the training data (over-fitting); (3) Testing data: once we have fully trained the model, we run a final dataset through the model as our ultimate test of the algorithm (see *Tables 1 and 2*). Paperspace's gradient ML Platform was used for training the models. The machine used was one with NVIDIA Quadra P5000 or P6000 GPU. We trained using an Adam optimisation.

## Detecting divisions from U-NetCellDivision outputs

After running a full video through our model in individual video clips, we have output masks with white circles in the same locations as the divisions (see *Figure 2A, B*). We detect the white circles using a Laplacian of Gaussian Filter (*Kong et al., 2013*). The deep learning model is very accurate at finding divisions when they occur, but sometimes mistakenly detects them a frame before and/or after the actual division happens. This may be expected as the video clips still look similar after being shifted by one timepoint. The white circles in the frames before and after are normally not as intense as the timepoint of the division, reflecting the weaker confidence of the model in identifying them. To ensure we do not double count divisions, these are suppressed with the brightest circle taken as the timepoint when a cell divides. We have built in some tolerance into our evaluation of the model. When the algorithm detects one of these divisions and has a brighter spot in a timepoint ±1 frame of our labelled data, we still count this as a correctly detected division.

## Orientation from U-NetOrientation outputs

To determine the orientation of the oval shape produced by the U-NetOrientation deep learning model, we calculated a second-rank tensor (which we call the $q$-tensor) for the output image that stores information about the orientation of the oval shapes.

$$\boldsymbol{q} = \frac{1}{A^2} \int_A \begin{pmatrix} \frac{1}{2}\left(x^2 - y^2\right) & xy \\ xy & \frac{1}{2}\left(y^2 - x^2\right) \end{pmatrix} dA$$

where $A$ is the area of the image and $dA = dxdy$. $\boldsymbol{q}$ can be rewritten as

$$\boldsymbol{q} = q_0 \begin{pmatrix} \cos 2\theta & \sin 2\theta \\ \sin 2\theta & -\cos 2\theta \end{pmatrix}$$

where $\theta$ is the orientation of the shape. To calculate the orientation of a division, we apply these equations to our output image from U-NetOrientation and extract $\theta$.

## Using Tissue Analzyer for segmentation

We use the watershed algorithm from Tissue Analzyer (*Etournay et al., 2016*), which is a plugin for ImageJ for segmentation both from the single focal plane data and from the output of the deep learning U-NetBoundary model. The 'Detect bonds V3' function was used to perform the segmentation. We found individual optimised settings for both single focal plane and U-NetBoundary output images. These were not the same settings, as the images are very different. To track cells after segmentation, we used the 'Track cell (static tissue)' algorithm. The U-NetBoundary outputs are resized back to 512 × 512 before being run through Tissue Analyzer.

## How to adapt this method for other cell division datasets

The models we have developed (optimised for *Drosophila* pupal wing epithelia) can be used on datasets from other systems. To be effective on a new tissue type, retraining will typically be needed. In our GitHub repository, we include the scripts for training new models (https://github.com/turleyjm/cell-division-dl-plugin; *Turley, 2024*). For best results, the user should load our model and weights, then train the model from this starting point (called transfer learning; *Howard and Gugger, 2020*). The user will also need to generate labelled data (as done in the 'Imaging and data processing' section). Once this has been done, the user can utilise the training scripts to teach the model. Other researchers may wish to vary the number of input channels to use longer/shorter videos or different numbers of fluorescent channels. This can easily be changed in the code for the model, with comments on the GitHub repository highlighting where alterations need to be made. Additionally, the image classifier model, which is currently converted to a U-Net (currently ResNet34), can also be replaced by a different classification network. This allows for different models to be incorporated.

## Wound, division density, and orientation measurements

The epithelial wound was located using the 2D focussed E-cadherin channel. The ImageJ plugin Trainable Weka Segmentation (a machine learning algorithm) is trained to find areas of the images that are tissue or non-tissue. Non-tissue could be either a wound or parts of the tissue that are above or below our frame of reference. The tissue/non-tissue binary masks are then hand-edited to remove errors (mostly around the edges of wounds where the images are particularly noisy due to debris). To calculate the division density, we sum the number of cell divisions divided by the area of tissue during a defined time period. We find the number of divisions from our deep learning model, and using the tissue/non-tissue binary masks, we know the area of tissue observed in the video. For measuring division density in relation to a wound, the mask could then be used to extract the wound. We then calculated the distance from the edge of a wound to the divisions using a distance transform (*Fabbri and Da, 2008*). Now we can find all the divisions in a band of a given radius and width. To quantify the density of divisions, we divide the number of divisions by the area of the band. Using both the distance transform and our tissue mask, we can work out the area of the tissue that is in each band. Once the wound has closed, we can no longer perform a distance transform using the wound edge, so we instead take the centre of the last timepoint before the wound closes. This point is the wound site and is where we take our distance transform from. As the tissue is still developing and moving, we track this point over time.

We track the tissue using the ImageJ plugin TrackMate (*Tinevez et al., 2017*), which tracks the nuclei of cells as they move together in the tissue. Unlike the mitotic nuclei, these nuclei are slow

moving, so trackable using a non-AI algorithm. By calculating the average velocity of the cells around the wound site, we can track this point and use this as our frame of reference to measure the distance from the wound site. The same method is used for unwounded tissue where we chose an arbitrary point as a 'virtual wound', which will flow with the local tissue. The starting point for the unwounded tissue is the centre of the image. This gives us our reference point to identify the bands we use for calculating the division density. We measure the orientation of division relative to the centroid of the polygon approximating the boundary of a wound (which we call the wound centre). The difference in angle between the vector from the wound centre to divisions and nomadic division vector is defined as the division orientation. Once the wound had closed, the wound centre point was used, whereas for the unwounded tissue, we used the 'virtual wound'.

## Division density correlation function

We calculate the division density in our system as follows: we image a 123.26 × 123.26 μm section of the tissue for 186 min taking an image every 2 min. This video is converted into a 3D ($x,y,t$) matrix of dimensions 124 × 124 × 89, whose components are 1 where there is a division and 0 otherwise. Thus, each component represents a 1 μm$^2$-2 min space–time slice. We defined the time of division as the moment that anaphase starts. We use only 89 (and not all 93) time slices because we have incomplete information about division at the beginning and end of the video.

We number each of the elements in the matrix $i \in [1, \ldots, N]$ where $N = 1,368,464$ is the number of elements. For the $i$th element, we define the mean mitosis density, $M^i(t, r)$, to be the division density in a space–time annular tube spatially centred at the point corresponding to the $i$th element, with spatial radius $(r - \delta r, r]$; temporally, the annular tube is of extent $(T - t, T - t + \delta t]$. Here, $T$ is the time corresponding to the $i$th element, $\delta r$ is 10 μm and $\delta t$ is 10 min; this is the size of our bins. Consequently, $M^i$ is defined for $t = 10, 20$ min, etc. and similarly for $r = 10, 20$ μm, etc. When calculating the density of a tube we take the number of divisions in the region and divide by the space–time volume, but we need to take into account the fact that often the annular tube will extend outside the microscope view. Therefore, we divide only by the volume that can be observed using the confocal. It is convenient to extend the definition of mean mitosis density also to $t = 0$ and $r = 0$. When $t = 0$, the annular tube has no temporal depth and is concerned only with the time $T$. Similarly, when $r = 0$, the annular tube becomes a line. When both are 0, the annular tube becomes a single point in space time. We define $M^i(0, 0) = 1$ if the $i$th element is a division and $M^i(0, 0) = 0$ otherwise.

We define the correlations between the divisions as:

$$\langle M(0,0) M(t,r) \rangle_c = \langle M(0,0) M(t,r) \rangle - \langle M(0,0) \rangle \langle M(t,r) \rangle.$$

The first term on the right hand side (RHS) is

$$\langle M(0,0) M(t,r) \rangle = \frac{1}{N} \sum_i^N M^i(0,0) M^i(t,r) = \frac{1}{N} \sum_{i \in d} M^i(t,r),$$

where $d$ is the subset of elements where $M^i(0,0) = 1$, that is, corresponds to a division. This means that this term is looking only at the densities around the divisions. The second term is

$$\langle M(0,0) \rangle = \frac{1}{N} \sum_i^N M^i(0,0) = \frac{N_d}{N},$$

where $N_d$ is the number of divisions in the video. The last term is

$$\langle M(t,r) \rangle = \frac{1}{N} \sum_i^N M^i(t,r) \approx \frac{1}{N_R} \sum_{i \in R} M^i(t,r).$$

Here, since the computation would take an extremely long time as there are $N = 1,368,464$ elements, we approximate it by randomly choosing a subset, $R$, of $N_R = 1000$ elements.

The resulting division density correlation function (*Figure 3B*) shows that there is a positive correlation in space and time, so $\langle M(0,0)M(t,r) \rangle > \langle M(0,0) \rangle \langle M(t,r) \rangle$ for $r < 40$m and $t < 50$. This means that if we find one mitotic event we are more likely to find others nearby and soon afterwards.

## Division orientation correlation function

We compute the orientation angle of each division using U-NetOrientation, and form the orientation vector:

$$\boldsymbol{p}\left(\theta\right) = \begin{pmatrix} \cos 2\theta \\ \sin 2\theta \end{pmatrix}.$$

Here, $2\theta$ is used since cell division orientation is nematic and we need $\boldsymbol{p}(\theta) = \boldsymbol{p}(\theta + \pi)$. To compare two division orientations, we take the dot product of the orientation vectors: 1 indicates that the divisions are aligned, $-1$ that they are perpendicular, and 0 that their orientations differ by $\frac{\pi}{4}$.

The division orientation correlation function is defined as,

$$T\left(t, r\right) = \left\langle \boldsymbol{p_i} \cdot \boldsymbol{p_j} \right\rangle$$

where $\langle \boldsymbol{p_i} \cdot \boldsymbol{p_j} \rangle$ is the mean dot product comparing the orientation of every pair of divisions within a radius $(r - \delta r, r]$ and $(t - \delta t, t]$ time from each other. This is calculated as explained above and shown in *Figure 4C*. Values of $T(t, r)$ close to 1 indicate highly aligned divisions, 0 no correlation and $-1$ anti-correlated.

## Acknowledgements

We would like to thank members of the Weavers, Martin, Chenchiah, and Liverpool groups for helpful discussion. We also thank the Wolfson Bioimaging Facility (Bristol, UK), particularly Stephen Cross, for help setting up pyimagej, for helpful conversations and sharing useful resources. We are grateful to the Drosophila research community, Flybase and the Bloomington Stock Centre (Indiana, US), for various fly lines/reagents. We thank Jack Dymond for helpful conversations and sharing useful resources. This research was funded by the MRC-GW4 DTP PhD programme (scholarship to JT) [MR/N013794/1, NE/W503174/1, studentship 2284082], a Wellcome Trust and Royal Society Sir Henry Dale Fellowship to HW [208762/Z/17/Z] and a Wellcome Trust Investigator Award to PM [217169/Z/19/Z], Eric and Wendy Schmidt AI in Science Postdoctoral Fellowship to JT. For the purpose of Open Access, the authors have applied a CC BY public copyright license to any Author Accepted Manuscript arising from this submission.

## Additional information

### Funding

| Funder | Grant reference number | Author |
| --- | --- | --- |
| Engineering and Physical Sciences Research Council | EP/R014604/1 | Jake Turley<br>Isaac V Chenchiah<br>Tanniemola B Liverpool |
| Engineering and Physical Sciences Research Council | EP/T031077/1 | Jake Turley<br>Isaac V Chenchiah<br>Tanniemola B Liverpool |
| Wellcome Trust | 10.35802/217169 | Paul Martin |
| Wellcome Trust / Royal Society | 10.35802/208762 | Helen Weavers |
| Medical Research Council | MR/N013794/1 | Jake Turley |
| Medical Research Council | NE/W503174/1 | Jake Turley |
| Medical Research Council | 2284082 | Jake Turley |

The funders had no role in study design, data collection, and interpretation, or the decision to submit the work for publication. For the purpose of Open Access, the authors have applied a CC BY public copyright license to any Author Accepted Manuscript version arising from this submission.

## Author contributions
Jake Turley, Conceptualization, Resources, Data curation, Software, Formal analysis, Validation, Investigation, Visualization, Methodology, Writing – original draft, Writing – review and editing; Isaac V Chenchiah, Conceptualization, Supervision, Methodology, Writing – original draft, Project administration, Writing – review and editing; Paul Martin, Helen Weavers, Conceptualization, Resources, Supervision, Methodology, Writing – original draft, Project administration, Writing – review and editing; Tanniemola B Liverpool, Conceptualization, Formal analysis, Supervision, Methodology, Writing – original draft, Project administration, Writing – review and editing

## Author ORCIDs
Jake Turley ⓘ https://orcid.org/0000-0001-8553-4367
Isaac V Chenchiah ⓘ https://orcid.org/0000-0002-8618-620X
Tanniemola B Liverpool ⓘ https://orcid.org/0000-0003-4376-5604
Helen Weavers ⓘ https://orcid.org/0000-0002-5383-6085

Reviewer #1 (Public review): https://doi.org/10.7554/eLife.87949.3.sa1
Reviewer #2 (Public review): https://doi.org/10.7554/eLife.87949.3.sa2
Author response https://doi.org/10.7554/eLife.87949.3.sa3

## Additional files

### Supplementary files
MDAR checklist

### Data availability
All data generated or analysed during this study has been deposited in Zenodo at this link https://zenodo.org/records/13819609.

The following dataset was generated:

| Author(s) | Year | Dataset title | Dataset URL | Database and Identifier |
|---|---|---|---|---|
| Turley J, Chenchiah IV, Martin P, Liverpool TB, Weavers H | 2024 | Deep learning for rapid analysis of cell divisions in vivo during epithelial morphogenesis and repair | https://doi.org/10.5281/zenodo.13819609 | Zenodo, 10.5281/zenodo.13819609 |

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
