## [Editor Report · eLife assessment]

In this **valuable** study, the authors use deep learning models to provide **solid** evidence that epithelial wounding triggers bursts of cell division at a characteristic distance away from the wound. The documentation provided by the authors should allow other scientists to readily apply these methods, which are particularly appropriate where unsupervised machine-learning algorithms have difficulties.

---

## [Referee Report · Reviewer #1 (Public review)]

The authors present a number of deep learning models to analyse the dynamics of epithelia. In this way they want to overcome the time-consuming manual analysis of such data and also remove a potential operator bias. Specifically, they set up models for identifying cell division events and cell division orientation. They apply these tools to the epithelium of the developing *Drosophila* pupal wing. They confirm a linear decrease of the division density with time and identify a burst of cell division after healing of a wound that they had induced earlier. These division events happen a characteristic time after and a characteristic distance away from the wound. These characteristic quantities depend on the size of the wound.

Strengths:

The methods developed in this work achieve the goals set by the authors and are a very helpful addition to the toolbox of developmental biologists. They could potentially be used on various developing epithelia. The evidence for the impact of wounds on cell division is compelling.

The methods presented in this work should prove to be very helpful for quantifying cell proliferation in epithelial tissues.

---

## [Referee Report · Reviewer #2 (Public review)]

In this manuscript, the authors propose a computational method based on deep convolutional neural networks (CNNs) to automatically detect cell divisions in two-dimensional fluorescence microscopy timelapse images. Three deep learning models are proposed to detect the timing of division, predict the division axis, and enhance cell boundary images to segment cells before and after division. Using this computational pipeline, the authors analyze the dynamics of cell divisions in the epithelium of the *Drosophila* pupal wing and find that a wound first induces a reduction in the frequency of division followed by a synchronised burst of cell divisions about 100 minutes after its induction.

Comments on revised version:

Regarding the Reviewer's 1 comment on the architecture details, I have now understood that the precise architecture (number/type of layers, activation functions, pooling operations, skip connections, upsampling choice...) might have remained relatively hidden to the authors themselves, as the U-net is built automatically by the fast.ai library from a given classical choice of encoder architecture (ResNet34 and ResNet101 here) to generate the decoder part and skip connections.

Regarding the Major point 1, I raised the question of the generalisation potential of the method. I do not think, for instance, that the optimal number of frames to use, nor the optimal choice of their time-shift with respect to the division time (t-n, t+m) (not systematically studied here) may be generic hyperparameters that can be directly transferred to another setting. This implies that the method proposed will necessarily require re-labeling, re-training and re-optimizing the hyperparameters which directly influence the network architecture for each new dataset imaged differently. This limits the generalisation of the method to other datasets, and this may be seen as in contrast to other tools developed in the field for other tasks such as cellpose for segmentation, which has proven a true potential for generalisation on various data modalities. I was hoping that the authors would try themselves testing the robustness of their method by re-imaging the same tissue with slightly different acquisition rate for instance, to give more weight to their work.

In this regard, and because the authors claimed to provide clear instructions on how to reuse their method or adapt it to a different context, I delved deeper into the code and, to my surprise, felt that we are far from the coding practice of what a well-documented and accessible tool should be.

To start with, one has to be relatively accustomed with Napari to understand how the plugin must be installed, as the only thing given is a pip install command (that could be typed in any terminal without installing the plugin for Napari, but has to be typed inside the Napari terminal, which is mentioned nowhere). Surprisingly, the plugin was not uploaded on Napari hub, nor on PyPI by the authors, so it is not searchable/findable directly, one has to go to the Github repository and install it manually. In that regard, no description was provided in the copy-pasted templated files associated to the napari hub, so exporting it to the hub would actually leave it undocumented.

Regarding now the python notebooks, one can fairly say that the "clear instructions" that were supposed to enlighten the code are really minimal. Only one notebook "trainingUNetCellDivision10.ipynb" has actually some comments, the other have (almost) none nor title to help the unskilled programmer delving into the script to guess what it should do. I doubt that a biologist who does not have a strong computational background will manage adapting the method to its own dataset (which seems to me unavoidable for the reasons mentioned above).

Finally regarding the data, none is shared publicly along with this manuscript/code, such that if one doesn't have a similar type of dataset - that must be first annotated in a similar manner - one cannot even test the networks/plugin for its own information. A common and necessary practice in the field - and possibly a longer lasting contribution of this work - could have been to provide the complete and annotated dataset that was used to train and test the artificial neural network. The basic reason is that a more performant, or more generalisable deep-learning model may be developed very soon after this one and for its performance to be fairly compared, it requires to be compared on the same dataset. Benchmarking and comparison of methods performance is at the core of computer vision and deep-learning.

---

## [Author Response]

The following is the authors’ response to the current reviews.

**Reviewer #1 (Public Review):**
The authors present a number of deep learning models to analyse the dynamics of epithelia. In this way they want to overcome the time-consuming manual analysis of such data and also remove a potential operator bias. Specifically, they set up models for identifying cell division events and cell division orientation. They apply these tools to the epithelium of the developing *Drosophila* pupal wing. They confirm a linear decrease of the division density with time and identify a burst of cell division after healing of a wound that they had induced earlier. These division events happen a characteristic time after and a characteristic distance away from the wound. These characteristic quantities depend on the size of the wound.Strengths:The methods developed in this work achieve the goals set by the authors and are a very helpful addition to the toolbox of developmental biologists. They could potentially be used on various developing epithelia. The evidence for the impact of wounds on cell division is compelling.The methods presented in this work should prove to be very helpful for quantifying cell proliferation in epithelial tissues.

We thank the reviewer for the positive comments!

**Reviewer #2 (Public Review):**
In this manuscript, the authors propose a computational method based on deep convolutional neural networks (CNNs) to automatically detect cell divisions in two-dimensional fluorescence microscopy timelapse images. Three deep learning models are proposed to detect the timing of division, predict the division axis, and enhance cell boundary images to segment cells before and after division. Using this computational pipeline, the authors analyze the dynamics of cell divisions in the epithelium of the *Drosophila* pupal wing and find that a wound first induces a reduction in the frequency of division followed by a synchronised burst of cell divisions about 100 minutes after its induction.Comments on revised version:Regarding the Reviewer's 1 comment on the architecture details, I have now understood that the precise architecture (number/type of layers, activation functions, pooling operations, skip connections, upsampling choice…) might have remained relatively hidden to the authors themselves, as the U-net is built automatically by the fast.ai library from a given classical choice of encoder architecture (ResNet34 and ResNet101 here) to generate the decoder part and skip connections.Regarding the Major point 1, I raised the question of the generalisation potential of the method. I do not think, for instance, that the optimal number of frames to use, nor the optimal choice of their time-shift with respect to the division time (t-n, t+m) (not systematically studied here) may be generic hyperparameters that can be directly transferred to another setting. This implies that the method proposed will necessarily require re-labeling, re-training and re-optimizing the hyperparameters which directly influence the network architecture for each new dataset imaged differently. This limits the generalisation of the method to other datasets, and this may be seen as in contrast to other tools developed in the field for other tasks such as cellpose for segmentation, which has proven a true potential for generalisation on various data modalities. I was hoping that the authors would try themselves testing the robustness of their method by re-imaging the same tissue with slightly different acquisition rate for instance, to give more weight to their work.

We thank the referee for the comments. Regarding this particular biological system, due to photobleaching over long imaging periods (and the availability of imaging systems during the project), we would have difficulty imaging at much higher rates than the 2 minute time frame we currently use. These limitations are true for many such systems, and it is rarely possible to rapidly image for long periods of time in real experiments. Given this upper limit in framerate, we could, in principle, sample this data at a lower framerate, by removing time points of the videos but this typically leads to worse results. With some pilot data, we have tried to use fewer time intervals for our analysis but they always gave worse results. We found we need to feed the maximum amount of information available into the model to get the best results (i.e. the fastest frame rate possible, given the data available). Our goal is to teach the neural net to identify dynamic space-time localised events from time lapse videos, in which the duration of an event is a key parameter. Our division events take 10 minutes or less to complete therefore we used 5 timepoints in the videos for the deep learning model. If we considered another system with dynamic events which have a duration T when we would use T/t timepoints where t is the minimum time interval (for our data t=2min). For example if we could image every minute we would use 10 timepoints. As discussed below, we do envision other users with different imaging setups and requirements may need to retrain the model for their own data and to help with this, we have now provided more detailed instructions how to do this (see later).

In this regard, and because the authors claimed to provide clear instructions on how to reuse their method or adapt it to a different context, I delved deeper into the code and, to my surprise, felt that we are far from the coding practice of what a well-documented and accessible tool should be.To start with, one has to be relatively accustomed with Napari to understand how the plugin must be installed, as the only thing given is a pip install command (that could be typed in any terminal without installing the plugin for Napari, but has to be typed inside the Napari terminal, which is mentioned nowhere). Surprisingly, the plugin was not uploaded on Napari hub, nor on PyPI by the authors, so it is not searchable/findable directly, one has to go to the Github repository and install it manually. In that regard, no description was provided in the copy-pasted templated files associated to the napari hub, so exporting it to the hub would actually leave it undocumented.

We thank the referee for suggesting the example of (DeXtrusion, Villars et al. 2023). We have endeavoured to produce similarly-detailed documentation for our tools. We now have clear instructions for installation requiring only minimal coding knowledge, and we have provided a user manual for the napari plug-in. This includes information on each of the options for using the model and the outputs they will produce. The plugin has been tested by several colleagues using both Windows and Mac operating systems.

Regarding now the python notebooks, one can fairly say that the "clear instructions" that were supposed to enlighten the code are really minimal. Only one notebook "trainingUNetCellDivision10.ipynb" has actually some comments, the other have (almost) none nor title to help the unskilled programmer delving into the script to guess what it should do. I doubt that a biologist who does not have a strong computational background will manage adapting the method to its own dataset (which seems to me unavoidable for the reasons mentioned above).

Within the README file, we have now included information on how to retrain the models with helpful links to deep learning tutorials (which, indeed, some of us have learnt from) for those new to deep learning. All Jupyter notebooks now include more comments explaining the models.

Finally regarding the data, none is shared publicly along with this manuscript/code, such that if one doesn't have a similar type of dataset - that must be first annotated in a similar manner - one cannot even test the networks/plugin for its own information. A common and necessary practice in the field - and possibly a longer lasting contribution of this work - could have been to provide the complete and annotated dataset that was used to train and test the artificial neural network. The basic reason is that a more performant, or more generalisable deep-learning model may be developed very soon after this one and for its performance to be fairly compared, it requires to be compared on the same dataset. Benchmarking and comparison of methods performance is at the core of computer vision and deep-learning.

We thank the referee for these comments. We have now uploaded all the data used to train the models and to test them, as well as all the data used in the analyses for the paper. This includes many videos that were not used for training but were analysed to generate the paper’s results. The link to these data sets is provided in our GitHub page (https://github.com/turleyjm/cell-division-dl-plugin/tree/main). In the folder for the data sets and in the GitHub repository, we have included the Jupyter notebooks used to train the models and these can be used for retraining. We have made our data publicly available at Zenodo dataset https://zenodo.org/records/10846684 (added to last paragraph of discussion). We have also included scripts that can be used to compare the model output with ground truth, including outputs highlighting false positives and false negatives. Together with these scripts, models can be compared and contrasted, both in general and in individual videos. Overall, we very much appreciate the reviewer’s advice, which has made the plugin much more user- friendly and, hopefully, easier for other groups to train their own models. Our contact details are provided, and we would be happy to advise any groups that would like to use our tools.

The following is the authors’ response to the original reviews.

**Reviewer #1 (Public Review):**
The authors present a number of deep-learning models to analyse the dynamics of epithelia. In this way, they want to overcome the time-consuming manual analysis of such data and also remove a potential operator bias. Specifically, they set up models for identifying cell division events and cell division orientation. They apply these tools to the epithelium of the developing *Drosophila* pupal wing. They confirm a linear decrease of the division density with time and identify a burst of cell division after the healing of a wound that they had induced earlier. These division events happen a characteristic time after and a characteristic distance away from the wound. These characteristic quantities depend on the size of the wound.Strength:The methods developed in this work achieve the goals set by the authors and are a very helpful addition to the toolbox of developmental biologists. They could potentially be used on various developing epithelia. The evidence for the impact of wounds on cell division is solid.Weakness:Some aspects of the deep-learning models remained unclear, and the authors might want to think about adding details. First of all, for readers not being familiar with deep-learning models, I would like to see more information about ResNet and U-Net, which are at the base of the new deep-learning models developed here. What is the structure of these networks?

We agree with the Reviewer and have included additional information on page 8 of the manuscript, outlining some background information about the architecture of ResNet and U-Net models.

How many parameters do you use?

We apologise for this omission and have now included the number of parameters and layers in each model in the methods section on page 25.

What is the difference between validating and testing the model? Do the corresponding data sets differ fundamentally?

The difference between ‘validating’ and ‘testing’ the model is validating data is used during training to determine whether the model is overfitting. If the model is performing well on the training data but not on the validating data, this a key signal the model is overfitting and changes will need to be made to the network/training method to prevent this. The testing data is used after all the training has been completed and is used to test the performance of the model on fresh data it has not been trained on. We have removed refence to the validating data in the main text to make it simpler and add this explanation to the methods. There is no fundamental (or experimental) difference between each of the labelled data sets; rather, they are collected from different biological samples. We have now included this information in the Methods text on page 24.

How did you assess the quality of the training data classification?

These data were generated and hand-labelled by an expert with many years of experience in identifying cell divisions in imaging data, to give the ground truth for the deep learning model.

**Reviewer #1 (Recommendations For The Authors):**
You repeatedly use 'new', 'novel' as well as 'surprising' and 'unexpected'. The latter are rather subjective and it is not clear based on what prior knowledge you make these statements. Unless indicated otherwise, it is understood that the results and methods are new, so you can delete these terms.

We have deleted these words, as suggested, for almost all cases.

p.4 "as expected" add a reference or explain why it is expected.

A reference has now been included in this section, as suggested.

p.4 "cell divisions decrease linearly with time" Only later (p.10) it turns out that you think about the density of cell divisions.

This has been changed to "cell division density decreases linearly with time".

p.5 "imagine is largely in one plane" while below "we generated a 3D z-stack" and above "our in vivo 3D image data" (p.4). Although these statements are not strictly contradictory, I still find them confusing. Eventually, you analyse a 2D image, so I would suggest that you refer to your in vivo data as being 2D.

We apologise for the confusion here; the imaging data was initially generated using 3D z-stacks but this 3D data is later converted to a 2D focused image, on which the deep learning analysis is performed. We are now more careful with the language in the text.

p.7 "We have overcome (...) the standard U-Net model" This paragraph remains rather cryptic to me. Maybe you can explain in two sentences what a U-Net is or state its main characteristics. Is it important to state which class you have used at this point? Similarly, what is the exact role of the ResNet model? What are its characteristics?

We have included more details on both the ResNet and U-Net models and how our model incorporates properties from them on Page 8.

p.8 Table 1 Where do I find it? Similarly, I could not find Table 2.

These were originally located in the supplemental information document, but have been moved to the main manuscript.

p.9 "developing tissue in normal homeostatic conditions" Aren't homeostatic and developing contradictory? In one case you maintain a state, in the other, it changes.

We agree with the Reviewer and have removed the word ‘homeostatic’.

p.9 "Develop additional models" I think 'models' refers to deep learning models, not to physical models of epithelial tissue development. Maybe you can clarify this?

Yes, this is correct; we have phrased this better in the text.

p.12 "median error" median difference to the manually acquired data?

Yes, and we have made this clearer in the text, too.

p.12 "we expected to observe a bias of division orientation along this axis" Can you justify the expectation? Elongated cells are not necessarily aligned with the direction of a uniaxially applied stress.

Although this is not always the case, we have now included additional references to previous work from other groups which demonstrated that wing epithelial cells do become elongated along the P/D axis in response to tension.

p.14 "a rather random orientation" Please, quantify.

The division orientations are quantified in Fig. 4F,G; we have now changed our description from ‘random’ to ‘unbiased’.

p.17 "The theories that must be developed will be statistical mechanical (stochastic) in nature" I do not understand. Statistical mechanics refers to systems at thermodynamic equilibrium, stochastic to processes that depend on, well, stochastic input.

We have clarified that we are referring to non-equilibrium statistical mechanics (the study of macroscopic systems far from equilibrium, a rich field of research with many open problems and applications in biology).

**Reviewer #2 (Public Review):**
In this manuscript, the authors propose a computational method based on deep convolutional neural networks (CNNs) to automatically detect cell divisions in two-dimensional fluorescence microscopy timelapse images. Three deep learning models are proposed to detect the timing of division, predict the division axis, and enhance cell boundary images to segment cells before and after division. Using this computational pipeline, the authors analyze the dynamics of cell divisions in the epithelium of the *Drosophila* pupal wing and find that a wound first induces a reduction in the frequency of division followed by a synchronised burst of cell divisions about 100 minutes after its induction.In general, novelty over previous work does not seem particularly important. From a methodological point of view, the models are based on generic architectures of convolutional neural networks, with minimal changes, and on ideas already explored in general. The authors seem to have missed much (most?) of the literature on the specific topic of detecting mitotic events in 2D timelapse images, which has been published in more specialized journals or Proceedings. (TPMAI, CCVPR etc., see references below). Even though the image modality or biological structure may be different (non-fluorescent images sometimes), I don't believe it makes a big difference. How the authors' approach compares to this previously published work is not discussed, which prevents me from objectively assessing the true contribution of this article from a methodological perspective.On the contrary, some competing works have proposed methods based on newer - and generally more efficient - architectures specifically designed to model temporal sequences (Phan 2018, Kitrungrotsakul 2019, 2021, Mao 2019, Shi 2020). These natural candidates (recurrent networks, long-short-term memory (LSTM) gated recurrent units (GRU), or even more recently transformers), coupled to CNNs are not even mentioned in the manuscript, although they have proved their generic superiority for inference tasks involving time series (Major point 2). Even though the original idea/trick of exploiting the different channels of RGB images to address the temporal aspect might seem smart in the first place - as it reduces the task of changing/testing a new architecture to a minimum - I guess that CNNs trained this way may not generalize very well to videos where the temporal resolution is changed slightly (Major point 1). This could be quite problematic as each new dataset acquired with a different temporal resolution or temperature may require manual relabeling and retraining of the network. In this perspective, recent alternatives (Phan 2018, Gilad 2019) have proposed unsupervised approaches, which could largely reduce the need for manual labeling of datasets.

We thank the reviewer for their constructive comments. Our goal is to develop a cell detection method that has a very high accuracy, which is critical for practical and effective application to biological problems. The algorithms need to be robust enough to cope with the difficult experimental systems we are interested in studying, which involve densely packed epithelial cells within in vivo tissues that are continuously developing, as well as repairing. In response to the above comments of the reviewer, we apologise for not including these important papers from the division detection and deep learning literature, which are now discussed in the Introduction (on page 4).

A key novelty of our approach is the use of multiple fluorescent channels to increase information for the model. As the referee points out, our method benefits from using and adapting existing highly effective architectures. Hence, we have been able to incorporate deeper models than some others have previously used. An additional novelty is using this same model architecture (retrained) to detect cell division orientation. For future practical use by us and other biologists, the models can easily be adapted and retrained to suit experimental conditions, including different multiple fluorescent channels or number of time points. Unsupervised approaches are very appealing due to the potential time saved compared to manual hand labelling of data. However, the accuracy of unsupervised models are currently much lower than that of supervised (as shown in Phan 2018) and most importantly well below the levels needed for practical use analysing inherently variable (and challenging) in vivo experimental data.

Regarding the other convolutional neural networks described in the manuscript:(1) The one proposed to predict the orientation of mitosis performs a regression task, predicting a probability for the division angle. The architecture, which must be different from a simple Unet, is not detailed anywhere, so the way it was designed is difficult to assess. It is unclear if it also performs mitosis detection, or if it is instead used to infer orientation once the timing and location of the division have been inferred by the previous network.

The neural network used for U-NetOrientation has the same architecture as U-NetCellDivision10 but has been retrained to complete a different task: finding division orientation. Our workflow is as follows: firstly, U-NetCellDivision10 is used to find cell divisions; secondly, U-NetOrientation is applied locally to determine the division orientation. These points have now been clarified in the main text on Page 14.

(2) The one proposed to improve the quality of cell boundary images before segmentation is nothing new, it has now become a classic step in segmentation, see for example Wolny et al. eLife 2020.

We have cited similar segmentation models in our paper and thank the referee for this additional one. We had made an improvement to the segmentation models, using GFP-tagged E-cadherin, a protein localised in a thin layer at the apical boundary of cells. So, while this is primarily a 2D segmentation problem, some additional information is available in the z-axis as the protein is visible in 2-3 separate z-slices. Hence, we supplied this 3-focal plane input to take advantage of the 3D nature of this signal. This approach has been made more explicit in the text (Pages 14, 15) and Figure (Fig. 2D).

As a side note, I found it a bit frustrating to realise that all the analysis was done in 2D while the original images are 3D z-stacks, so a lot of the 3D information had to be compressed and has not been used. A novelty, in my opinion, could have resided in the generalisation to 3D of the deep-learning approaches previously proposed in that context, which are exclusively 2D, in particular, to predict the orientation of the division.

Our experimental system is a relatively flat 2D tissue with the orientation of the cell divisions consistently in the xy-plane. Hence, a 2D analysis is most appropriate for this system. With the successful application of the 2D methods already achieving high accuracy, we envision that extension to 3D would only offer a slight increase in effectiveness as these measurements have little room for improvement. Therefore, we did not extend the method to 3D here. However, of course, this is the next natural step in our research as 3D models would be essential for studying 3D tissues; such 3D models will be computationally more expensive to analyse and more challenging to hand label.

Concerning the biological application of the proposed methods, I found the results interesting, showing the potential of such a method to automatise mitosis quantification for a particular biological question of interest, here wound healing. However, the deep learning methods/applications that are put forward as the central point of the manuscript are not particularly original.

We thank the referee for their constructive comments. Our aim was not only to show the accuracy of our models but also to show how they might be useful to biologists for automated analysis of large datasets, which is a—if not the—bottleneck for many imaging experiments. The ability to process large datasets will improve robustness of results, as well as allow additional hypotheses to be tested. Our study also demonstrated that these models can cope with real in vivo experiments where additional complications such as progressive development, tissue wounding and inflammation must be accounted for.

Major point 1: generalisation potential of the proposed method.The neural network model proposed for mitosis detection relies on a 2D convolutional neural network (CNN), more specifically on the Unet architecture, which has become widespread for the analysis of biology and medical images. The strategy proposed here exploits the fact that the input of such an architecture is natively composed of several channels (originally 3 to handle the 3 RGB channels, which is actually a holdover from computer vision, since most medical/biological images are gray images with a single channel), to directly feed the network with 3 successive images of a timelapse at a time. This idea is, in itself, interesting because no modification of the original architecture had to be carried out. The latest 10-channel model (U-NetCellDivision10), which includes more channels for better performance, required minimal modification to the original U-Net architecture but also simultaneous imaging of cadherin in addition to histone markers, which may not be a generic solution.

We believe we have provided a general approach for practical use by biologists that can be applied to a range of experimental data, whether that is based on varying numbers of fluorescent channels and/or timepoints. We envisioned that experimental biologists are likely to have several different parameters permissible for measurement based on their specific experimental conditions e.g., different fluorescently labelled proteins (e.g. tubulin) and/or time frames. To accommodate this, we have made it easy and clear in the code on GitHub how these changes can be made. While the model may need some alterations and retraining, the method itself is a generic solution as the same principles apply to very widely used fluorescent imaging techniques.

Since CNN-based methods accept only fixed-size vectors (fixed image size and fixed channel number) as input (and output), the length or time resolution of the extracted sequences should not vary from one experience to another. As such, the method proposed here may lack generalization capabilities, as it would have to be retrained for each experiment with a slightly different temporal resolution. The paper should have compared results with slightly different temporal resolutions to assess its inference robustness toward fluctuations in division speed.

If multiple temporal resolutions are required for a set of experiments, we envision that the model could be trained over a range of these different temporal resolutions. Of course, the temporal resolution, which requires the largest vector would be chosen as the model's fixed number of input channels. Given the depth of the models used and the potential to easily increase this by replacing resnet34 with resnet50 or resnet101 the model would likely be able to cope with this, although we have not specifically tested this. (page 27)

Another approach (not discussed) consists in directly convolving several temporal frames using a 3D CNN (2D+time) instead of a 2D, in order to detect a temporal event. Such an idea shares some similarities with the proposed approach, although in this previous work (Ji et al. TPAMI 2012 and for split detection Nie et al. CCVPR 2016) convolution is performed spatio-temporally, which may present advantages. How does the authors' method compare to such an (also very simple) approach?

We thank the Reviewer for this insightful comment. The text now discusses this (on Pages 8 and 17). Key differences between the models include our incorporation of multiple light channels and the use of much deeper models. We suggest that our method allows for an easy and natural extension to use deeper models for even more demanding tasks e.g. distinguishing between healthy and defective divisions. We also tested our method with ‘difficult conditions’ such as when a wound is present; despite the challenges imposed by the wound (including the discussed reduction in fluorescent intensities near the wound edge), we achieved higher accuracy compared to Nie et al. (accuracy of 78.5% compared to our F1 score of 0.964) using a low-density in vitro system.

Major point 2: innovatory nature of the proposed method.The authors' idea of exploiting existing channels in the input vector to feed successive frames is interesting, but the natural choice in deep learning for manipulating time series is to use recurrent networks or their newer and more stable variants (LSTM, GRU, attention networks, or transformers). Several papers exploiting such approaches have been proposed for the mitotic division detection task, but they are not mentioned or discussed in this manuscript: Phan et al. 2018, Mao et al. 2019, Kitrungrotaskul et al. 2019, She et al 2020.An obvious advantage of an LSTM architecture combined with CNN is that it is able to address variable length inputs, therefore time sequences of different lengths, whereas a CNN alone can only be fed with an input of fixed size.

LSTM architectures may produce similar accuracy to the models we employ in our study, however due to the high degree of accuracy we already achieve with our methods, it is hard to see how they would improve the understanding of the biology of wound healing that we have uncovered. Hence, they may provide an alternative way to achieve similar results from analyses of our data. It would also be interesting to see how LTSM architectures would cope with the noisy and difficult wounded data that we have analysed. We agree with the referee that these alternate models could allow an easier inclusion of difference temporal differences in division time (see discussion on Page 20). Nevertheless, we imagine that after selecting a sufficiently large input time/ fluorescent channel input, biologists could likely train our model to cope with a range of division lengths.

Another advantage of some of these approaches is that they rely on unsupervised learning, which can avoid the tedious relabeling of data (Phan et al. 2018, Gilad et al. 2019).

While these are very interesting ideas, we believe these unsupervised methods would struggle under the challenging conditions within ours and others experimental imaging data. The epithelial tissue examined in the present study possesses a particularly high density of cells with overlapping nuclei compared to the other experimental systems these unsupervised methods have been tested on. Another potential problem with these unsupervised methods is the difficulty in distinguishing dynamic debris and immune cells from mitotic cells. Once again despite our experimental data being more complex and difficult, our methods perform better than other methods designed for simpler systems as in Phan et al. 2018 and Gilad et al. 2019; for example, analysis performed on lower density in vitro and unwounded tissues gave best F1 scores for a single video was 0.768 and 0.829 for unsupervised and supervised respectively (Phan et al. 2018). We envision that having an F1 score above 0.9 (and preferably above 0.95), would be crucial for practical use by biologists, hence we believe supervision is currently still required. We expect that retraining our models for use in other experimental contexts will require smaller hand labelled datasets, as they will be able to take advantage of transfer learning (see discussion on Page 4).

References :

We have included these additional references in the revised version of our Manuscript.

Ji, S., Xu, W., Yang, M., & Yu, K. (2012). 3D convolutional neural networks for human action recognition. IEEE transactions on pattern analysis and machine intelligence, 35(1), 221-231. >6000 citations

Nie, W. Z., Li, W. H., Liu, A. A., Hao, T., & Su, Y. T. (2016). 3D convolutional networks-based mitotic event detection in time-lapse phase contrast microscopy image sequences of stem cell populations. In Proceedings of the IEEE Conference on Computer Vision and Pattern Recognition Workshops (pp. 55-62).

Phan, H. T. H., Kumar, A., Feng, D., Fulham, M., & Kim, J. (2018). Unsupervised two-path neural network for cell event detection and classification using spatiotemporal patterns. IEEE Transactions on Medical Imaging, 38(6), 1477-1487.

Gilad, T., Reyes, J., Chen, J. Y., Lahav, G., & Riklin Raviv, T. (2019). Fully unsupervised symmetry-based mitosis detection in time-lapse cell microscopy. Bioinformatics, 35(15), 2644-2653.

Mao, Y., Han, L., & Yin, Z. (2019). Cell mitosis event analysis in phase contrast microscopy images using deep learning. Medical image analysis, 57, 32-43.

Kitrungrotsakul, T., Han, X. H., Iwamoto, Y., Takemoto, S., Yokota, H., Ipponjima, S., ... & Chen, Y. W. (2019). A cascade of 2.5 D CNN and bidirectional CLSTM network for mitotic cell detection in 4D microscopy image. IEEE/ACM transactions on computational biology and bioinformatics, 18(2), 396-404.

Shi, J., Xin, Y., Xu, B., Lu, M., & Cong, J. (2020, November). A Deep Framework for Cell Mitosis Detection in Microscopy Images. In 2020 16th International Conference on Computational Intelligence and Security (CIS) (pp. 100-103). IEEE.

Wolny, A., Cerrone, L., Vijayan, A., Tofanelli, R., Barro, A. V., Louveaux, M., ... & Kreshuk, A. (2020). Accurate and versatile 3D segmentation of plant tissues at cellular resolution. Elife, 9, e57613.